# TRUNCATED DIFFUSION PROBABILISTIC MODELS AND DIFFUSION-BASED ADVERSARIAL AUTO-ENCODERS

**Huangjie Zheng[1,2] & Pengcheng He[2] & Weizhu Chen[2] & Mingyuan Zhou[1]**
The University of Texas at Austin[1], Microsoft Azure AI[2]
`huangjie.zheng@utexas.edu, penhe@microsoft.com,`
`wzchen@microsoft.com, mingyuan.zhou@mccombs.utexas.edu`

## ABSTRACT

Employing a forward diffusion chain to gradually map the data to a noise distribution, diffusion-based generative models learn how to generate the data by inferring a reverse diffusion chain. However, this approach is slow and costly because it needs many forward and reverse steps. We propose a faster and cheaper approach that adds noise not until the data become pure random noise, but until they reach a hidden noisy-data distribution that we can confidently learn. Then, we use fewer reverse steps to generate data by starting from this hidden distribution that is made similar to the noisy data. We reveal that the proposed model can be cast as an adversarial auto-encoder empowered by both the diffusion process and a learnable implicit prior. Experimental results show even with a significantly smaller number of reverse diffusion steps, the proposed truncated diffusion probabilistic models can provide consistent improvements over the non-truncated ones in terms of performance in both unconditional and text-guided image generations.

## 1 INTRODUCTION

Generating photo-realistic images with probabilistic models is a challenging and important task in machine learning and computer vision, with many potential applications in data augmentation, image editing, style transfer, *etc*. Recently, a new class of image generative models based on diffusion processes (Sohl-Dickstein et al., 2015) has achieved remarkable results on various commonly used image generation benchmarks (Song & Ermon, 2019; Ho et al., 2020; Song & Ermon, 2020; Song et al., 2021b; Dhariwal & Nichol, 2021), surpassing many existing deep generative models, such as autoregressive models (van den Oord et al., 2016), variational auto-encoders (VAEs) (Kingma & Welling, 2013; Rezende et al., 2014; van den Oord et al., 2017; Razavi et al., 2019), and generative adversarial networks (GANs) (Goodfellow et al., 2014; Radford et al., 2015; Arjovsky et al., 2017; Miyato et al., 2018; Brock et al., 2019; Karras et al., 2019; 2020b).

This new modeling class, which includes both score-based and diffusion-based generative models, uses noise injection to gradually corrupt the data distribution into a simple noise distribution that can be easily sampled from, and then uses a denoising network to reverse the noise injection to generate photo-realistic images. From the perspective of score matching (Hyvärinen & Dayan, 2005; Vincent, 2011) and Langevin dynamics (Neal, 2011; Welling & Teh, 2011), the denoising network is trained by matching the score function, which is the gradient of the log-density of the data, of the corrupted data distribution and that of the generator distribution at different noise levels (Song & Ermon, 2019). This training objective can also be formulated under diffusion-based generative models (Sohl-Dickstein et al., 2015; Ho et al., 2020). These two types of models have been further unified by Song et al. (2021b) under the framework of discretized stochastic differential equations.

Despite their impressive performance, diffusion-based (or score-based) generative models suffer from high computational costs, both in training and sampling. This is because they need to perform a large number of diffusion steps, typically hundreds or thousands, to ensure that the noise injection is small enough at each step to make the assumption that both the diffusion and denoising processes have the Gaussian form hold in the limit of small diffusion rate (Feller, 1949; Sohl-Dickstein et al., 2015). In other words, when the number of diffusion steps is small or when the rate is large, the Gaussian assumption may not hold well, and the model may not be able to capture the true score function

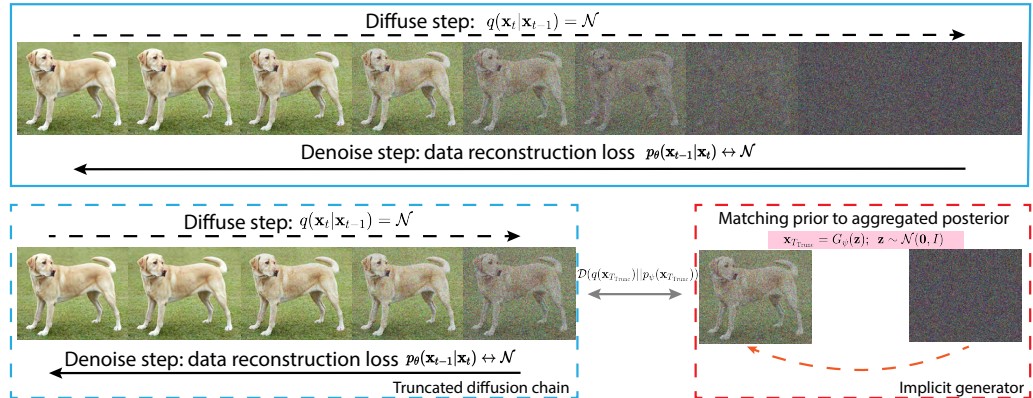

Figure 1: (*Best viewed in color*) An illustrative depiction of diffusion models and our truncated diffusion models. **Top**: The conventional denoising diffusion models add Gaussian noise gradually with a large number of time steps, where the true posterior can be kept close to Gaussian and hence easy to fit with denoising (score-matching) loss (marked in a solid blue box). **Bottom**: Truncated diffusion models truncate the diffusion chain to keep its first few steps and small diffusion segment (marked in the dashed blue box). This truncated diffusion chain can still be learned with previous denoising methods. Meanwhile, as the left part is truncated, the Gaussian prior $p(\mathbf{x}_T)$ will have a large gap to the truncated point $q(\mathbf{x}_T \mid \mathbf{x}_0)$, which is bridged with an implicit generative distribution $p_\psi(\mathbf{x}_T) = \int p_\psi(\mathbf{x}_T|\mathbf{z})p(\mathbf{z})d\mathbf{z}$ (marked in dashed red box).

of the data. Therefore, previous works have tried to reduce the number of diffusion steps by using non-Markovian reverse processes (Song et al., 2020; Kong & Ping, 2021), adaptive noise scheduling (San-Roman et al., 2021; Kingma et al., 2021), knowledge distillation (Luhman & Luhman, 2021; Salimans & Ho, 2022), diffusing in a lower-dimension latent space (Rombach et al., 2022), *etc.*, but they still cannot achieve significant speedup without sacrificing the generation quality.

In this paper, we propose a novel way to shorten the diffusion trajectory by learning an implicit distribution to start the reverse diffusion process, instead of relying on a tractable noise distribution. We call our method truncated diffusion probabilistic modeling (TDPM), which is based on the idea of truncating the forward diffusion chain of an existing diffusion model, such as the denoising diffusion probabilistic model (DDPM) of Ho et al. (2020). To significantly accelerate diffusion-based text-to-image generation, we also introduce the truncated latent diffusion model (TLDM), which truncates the diffusion chain of the latent diffusion model (LDM) of Rombach et al. (2022). We note LDM is the latent text-to-image diffusion model behind `Stable Diffusion`, an open-source project that provides state-of-the-art performance in generating photo-realistic images given text input. By truncating the chain, we can reduce the number of diffusion steps to an arbitrary level, but at the same time, we also lose the tractability of the distribution at the end of the chain. Therefore, we need to learn an implicit generative distribution that can approximate this distribution and provide the initial samples for the reverse diffusion process. We show that this implicit generative distribution can be implemented in different ways, such as using a separate generator network or reusing the denoising network. The former option has more flexibility and can improve the generation quality, while the latter option has no additional parameters and can achieve comparable results.

We reveal that DDPM and VAE have a similar relationship as TDPM and adversarial auto-encoder (AAE, Makhzani et al. (2015)). Specifically, DDPM is like a VAE with a fixed encoder and a learnable decoder that use a diffusion process, and a predefined prior. TDPM is like an AAE with a fixed encoder and a learnable decoder that use a truncated diffusion process, and a learnable implicit prior.

Our truncation method has several advantages when we use it to modify DDPM for generating images without text guidance or LDM for generating images with text guidance. First, it can generate samples much faster by using fewer diffusion steps, without sacrificing or even enhancing the generation quality. Second, it can exploit the cooperation between the implicit model and the diffusion model, as the diffusion model helps the implicit model train by providing noisy data samples, and the implicit model helps the diffusion model reverse by providing better initial samples. Third, it can adapt the truncation level to balance the generation quality and efficiency, depending on the data complexity and the computational resources. For generating images with text guidance, our method can speed up the generation significantly and make it suitable for real-time processing: while LDM takes the time to generate one photo-realistic image, our TLDM can generate more than 50 such images.

The main contributions of our paper are as follows:

- We introduce TDPM, a new diffusion-based generative model that can shorten the diffusion trajectory by learning an implicit distribution to start the reverse diffusion process, and demonstrate that the learning of the implicit distribution can be achieved in various ways. We further introduce TLDM to significantly accelerate diffusion-based text-to-image generation.
- We show TDPM can be formulated as a diffusion-based AAE.
- We show that the implicit distribution can be realized by reusing the denoising network for the reverse diffusion process, which can reduce the reverse diffusion steps by orders of magnitude without adding any extra parameters and with comparable generation quality.
- We reveal the synergy between the implicit model and the diffusion model, as the diffusion process can simplify the training of the implicit model like GANs, and the implicit model can speed up the reverse diffusion process of the diffusion model.
- We show that both TDPM and TLDM can adapt the truncation level, according to the data complexity and the computational resources, to achieve a good balance between the generation quality and the efficiency.

## 2 PRELIMINARIES ON DIFFUSION MODELS

In Gaussian diffusion models (Sohl-Dickstein et al., 2015; Ho et al., 2020), starting from the data distribution $\mathbf{x}_0 \sim q(\mathbf{x}_0)$, a pre-defined forward diffusion process $q_t$ produces auxiliary variables $\mathbf{x}_{t=1:T}$ by gradually adding Gaussian noise, with variance $\beta_t \in (0, 1)$ at time $t$, as follows:

$$q(\mathbf{x}_1, ..., \mathbf{x}_T \,|\, \mathbf{x}_0) \coloneqq \prod_{t=1}^{T} q(\mathbf{x}_t \,|\, \mathbf{x}_{t-1}), \quad q(\mathbf{x}_t \,|\, \mathbf{x}_{t-1}) \coloneqq \mathcal{N}(\mathbf{x}_t; \sqrt{1 - \beta_t}\mathbf{x}_{t-1}, \beta_t \boldsymbol{I}). \quad (1)$$

With the limit of small diffusion rate (*i.e.*, $\beta_t$ is kept sufficiently small), the reverse distribution $q(\mathbf{x}_{t-1} \,|\, \mathbf{x}_t)$ also follows a Gaussian distribution (Feller, 1949; Sohl-Dickstein et al., 2015) and can be approximated using a neural network parameterized Gaussian distribution $p_\theta$ as:

$$p_\theta(\mathbf{x}_{t-1} \,|\, \mathbf{x}_t) \coloneqq \mathcal{N}(\mathbf{x}_{t-1}; \mu_\theta(\mathbf{x}_t, t), \Sigma_\theta(\mathbf{x}_t, t)). \quad (2)$$

Moreover, with a sufficiently large $T$, the outcome of the diffusion chain $\mathbf{x}_T$ will follow an isotropic Gaussian distribution. Thus, with the pre-defined forward (inference) diffusion process and the learned reverse (generative) diffusion process, we can sample from $\mathbf{x}_T \sim \mathcal{N}(\mathbf{0}, \boldsymbol{I})$ and run the diffusion process in reverse to get a sample from the data distribution $q(\mathbf{x}_0)$.

Under the variational inference (Kingma & Welling, 2013; Blei et al., 2017) framework, viewing $q(\mathbf{x}_1, ..., \mathbf{x}_T \,|\, \mathbf{x}_0)$ in (1) as the inference network, we can use the evidence lower bound (ELBO) as our learning objective. Following previous works (Sohl-Dickstein et al., 2015; Ho et al., 2020), the negative ELBO of a diffusion probabilistic model, parameterized by $\theta$, can be expressed as

$$\mathcal{L}_{\mathrm{ELBO}}(\theta) \coloneqq \mathcal{L}_0(\theta) + \sum_{t=2}^{T} \mathcal{L}_{t-1}(\theta) + \mathcal{L}_T, \quad \mathcal{L}_0(\theta) \coloneqq \mathbb{E}_{q(\mathbf{x}_0)} \mathbb{E}_{q(\mathbf{x}_1 \,|\, \mathbf{x}_0)} \left[ -\log p_\theta(\mathbf{x}_0 \,|\, \mathbf{x}_1) \right], \quad (3)$$

$$\mathcal{L}_{t-1}(\theta) \coloneqq \mathbb{E}_{q(\mathbf{x}_0)} \mathbb{E}_{q(\mathbf{x}_t \,|\, \mathbf{x}_0)} [D_{\mathrm{KL}}\left( q(\mathbf{x}_{t-1} \,|\, \mathbf{x}_t, \mathbf{x}_0) || p_\theta(\mathbf{x}_{t-1} \,|\, \mathbf{x}_t) \right)], \quad t \in \{2, \ldots, T\} \quad (4)$$

$$\mathcal{L}_T \coloneqq \mathbb{E}_{q(\mathbf{x}_0)} [D_{\mathrm{KL}}\left( q(\mathbf{x}_T \,|\, \mathbf{x}_0) || p(\mathbf{x}_T) \right)], \quad (5)$$

where $D_{\mathrm{KL}}(q||p) = \mathbb{E}_q[\log q - \log p]$ denotes the Kullback–Leibler (KL) divergence from distributions $p$ to $q$. Generally speaking, diffusion probabilistic models assume the number of diffusion steps $T$ to be sufficiently large to satisfy two conditions: 1) the reverse distribution at each denoising step can be fitted with a Gaussian denoising generator $p_\theta(\mathbf{x}_{t-1}|\mathbf{x}_t)$; 2) with a sufficiently small diffusion rate $\beta_t$, the long forward diffusion process will successfully corrupt the data, making $q(\mathbf{x}_T \,|\, \mathbf{x}_0) \approx \mathcal{N}(\mathbf{0}, \boldsymbol{I})$, and hence approximately $L_T$ becomes zero and depends on neither $\mathbf{x}_0$ nor $\theta$.

**What happens if $T$ is insufficiently large?** Given a non-Gaussian data distribution $q(\mathbf{x}_0)$, when the number of denoising steps is reduced, the true posterior $q(\mathbf{x}_{t-1} \,|\, \mathbf{x}_t)$ is not Gaussian and usually intractable (Feller, 1949), resulting in new challenges to current diffusion models. As noted in Xiao et al. (2022), when $\beta_t$ is not sufficiently small, the diffusion step becomes larger and the denoising distribution can be multi-modal and hence too complex to be well fitted by Gaussian. The authors propose to define $p_\theta(\mathbf{x}_{t-1} \,|\, \mathbf{x}_t)$ with an implicit generator and substitute the ELBO with

$$\min_\theta \sum_{t \geq 1} \mathbb{E}_{q(t)} \left[ D_{\mathrm{adv}}(q(\mathbf{x}_{t-1} \,|\, \mathbf{x}_t) || p_\theta(\mathbf{x}_{t-1} \,|\, \mathbf{x}_t)) \right], \quad (6)$$

where $D_{\text{adv}}$ represents a statistical distance that relies on an adversarial training setup. This modified objective can be minimized by leveraging the power of conditional GANs in fitting implicit multi-modal distributions (Arjovsky et al., 2017; Goodfellow et al., 2014; Nowozin et al., 2016). While the concept of diffusion has been used, the proposed models in Xiao et al. (2022) are shown to work the best only when the number of diffusion steps is limited to be as few as four, and start to exhibit deteriorated performance when further increasing that number.

## 3 Truncated diffusion and adversarial auto-encoding

We first introduce the idea of accelerating both the training and generation of diffusion models by truncating the diffusion chains and describe the technical challenges. We then develop the objective function and training algorithm for TDPM. We further reveal TDPM can also be formulated as an AAE (Makhzani et al., 2015)) empowered by diffusion models. While DDPM can be considered as a hierarchical version of a variational auto-encoder (VAE) with a fixed multi-stochastic-layer encoder, our derivation shows that TDPM can be considered as a hierarchical version of an AAE with a fixed multi-stochastic-layer encoder but a learnable implicit prior.

### 3.1 Motivation and technical challenges

We propose a novel method called TDPM to speed up the diffusion process and the generative model. The main idea is to shorten the forward diffusion chain that transforms the data into Gaussian noise, and use a learned implicit distribution to sample the starting point of the reverse diffusion chain that reconstructs the data. To be more precise, we adopt the DDPM framework that defines a variance schedule $\{\beta_1, \beta_2, ..., \beta_T\}$, which controls the amount of noise added at each step of the forward diffusion process. The forward process has a simple analytical form as a Gaussian distribution:

$$q(\mathbf{x}_t \,|\, \mathbf{x}_0) = \mathcal{N}(\sqrt{\bar{\alpha}_t}\mathbf{x}_0, (1 - \bar{\alpha}_t)I); \quad \bar{\alpha}_t = \prod_{i=1}^{t} \alpha_i, \ \alpha_i = 1 - \beta_i.$$

Here, $\mathbf{x}_t$ is the noisy version of the data $\mathbf{x}_0$ at step $t$, and $\bar{\alpha}_t$ is the cumulative product of the diffusion coefficients $\alpha_i$. The forward chain of length $T$ is designed to be long enough to make the data distribution indistinguishable from Gaussian noise $\mathcal{N}(\mathbf{0}, \boldsymbol{I})$. However, a long forward chain also implies a high computational cost for the reverse process, which uses a learned neural network to predict the conditional distribution of the clean data given the noisy one at each step.

The proposed TDPM cuts off the last part of the forward chain and only keeps the first $T_{\text{trunc}}$ steps $\{\beta_1, \beta_2, ..., \beta_{T_{\text{trunc}}}\} \subset \{\beta_1, \beta_2, ..., \beta_T\}$. We choose $T_{\text{trunc}}$ to be much smaller than $T$ so that we can save a lot of computation time in generation. The benefit of this truncation is illustrated in Figure 1, where the bottom row shows the truncated diffusion chain. We can see that the data are only partially corrupted by noise and still retain some features of the original data. This means that we can recover the data more easily and accurately by applying a few Gaussian denoising steps from the corrupted data. Moreover, we do not change the diffusion rates $\beta_t$ for the first $T_{\text{trunc}}$ steps, so we do not compromise the quality of the forward and reverse processes between time 0 and $T_{\text{trunc}}$.

However, truncating the forward chain also introduces a new challenge for the reverse process. Unlike the original chain, where the starting point of the reverse process is $\mathbf{x}_T \sim \mathcal{N}(\mathbf{0}, \boldsymbol{I})$, the truncated chain has an unknown distribution of the corrupted data at step $T_{\text{trunc}}$. This makes it difficult to sample from this distribution and initiate the reverse process. To overcome this challenge, we introduce an implicit generative model that approximates the distribution of the corrupted data by minimizing a divergence measure between the implicit and the true noisy distributions at step $T_{\text{trunc}}$. This way, we can use the implicit model to sample the starting point of the reverse process and then apply the learned denoising network to generate the data.

### 3.2 Hand-crafted TDPM objective function

Mathematically, recall that the DDPM loss in (3) consists of three terms: $\mathcal{L}_0$, $\sum_{t=2}^{T} \mathcal{L}_{t-1}$, and $\mathcal{L}_T$. The training objective of a conventional diffusion model focuses on terms $\sum_{t=2}^{T} \mathcal{L}_{t-1}$ and $\mathcal{L}_0$. It assumes $\mathcal{L}_T$ does not depend on any parameter and will be close to zero by carefully pre-defining the forward noising process such that $q(\mathbf{x}_T \,|\, \mathbf{x}_0) \approx p(\mathbf{x}_T) = \mathcal{N}(\mathbf{0}, \boldsymbol{I})$.

When the diffusion chains are truncated at time $T_{\text{trunc}} \ll T$, the forward diffusion ends at time $T_{\text{trunc}}$, where the marginal distribution of the forward diffusion-corrupted data can be expressed as

$$q(\mathbf{x}_{T_{\text{trunc}}}) := \int q(\mathbf{x}_{T_{\text{trunc}}} \,|\, \mathbf{x}_0) p(\mathbf{x}_0) d\mathbf{x}_0, \tag{7}$$

which takes a semi-implicit form (Yin & Zhou, 2018) whose density function is often intractable. To reverse this truncated forward diffusion chain, we can no longer start the reverse diffusion chain from a known distribution such as $\mathcal{N}(\mathbf{0}, \boldsymbol{I})$. To this end, we propose TDPM that starts the reverse chain at time $T_{\text{trunc}}$ from $p_\psi(\mathbf{x}_{T_{\text{trunc}}})$, an implicit distribution parameterized by $\psi$. We match $p_\psi(\mathbf{x}_{T_{\text{trunc}}})$ to $q(\mathbf{x}_{T_{\text{trunc}}})$ via a loss term as $\tilde{\mathcal{L}}_{T_{\text{trunc}}} := \mathcal{D}\left(q(\mathbf{x}_{T_{\text{trunc}}})||p_\psi(\mathbf{x}_{T_{\text{trunc}}})\right)$, where $\mathcal{D}(q||p)$ is a statistical distance between distributions $q$ and $p$, such as the Jensen–Shannon divergence and Wasserstein distance. As we keep all the diffusion steps before time $T_{\text{trunc}}$ in TDPM the same as those in DDPM, we combine $\tilde{\mathcal{L}}_{T_{\text{trunc}}}$ with all the loss terms of DDPM before time $T_{\text{trunc}}$ in (3) to define the TDPM loss as

$$\mathcal{L}_{\text{TDPM}} := \textstyle\sum_{t=1}^{T_{\text{trunc}}} \mathcal{L}_{t-1}(\theta) + \tilde{\mathcal{L}}_{T_{\text{trunc}}}(\psi), \quad \tilde{\mathcal{L}}_{T_{\text{trunc}}}(\psi) := \mathcal{D}\left(q(\mathbf{x}_{T_{\text{trunc}}})||p_\psi(\mathbf{x}_{T_{\text{trunc}}})\right), \tag{8}$$

We note while in general $p_\psi(\mathbf{x}_{T_{\text{trunc}}})$ in TDPM is intractable, we can employ a deep neural network-based generator $G_\psi$ to generate a random sample in a single step via

$$\mathbf{x}_{T_{\text{trunc}}} = G_\psi(\mathbf{z}), \ \mathbf{z} \sim \mathcal{N}(\mathbf{0}, \boldsymbol{I}). \tag{9}$$

We will discuss later that we may simply let $\psi = \theta$ to avoid adding more parameters.

### 3.3 TDPM AS DIFFUSION-BASED ADVERSARIAL AUTO-ENCODER

Following the terminology of AAE, let us define the prior as $p_\psi(\mathbf{x}_{T_{\text{trunc}}})$, the decoder (likelihood) as

$$p_\theta(\mathbf{x}_0 \,|\, \mathbf{x}_{T_{\text{trunc}}}) := \int \ldots \int \left[ \textstyle\prod_{t=1}^{T_{\text{trunc}}} p_\theta(\mathbf{x}_{t-1} \,|\, \mathbf{x}_t) \right] d\mathbf{x}_{T_{\text{trunc}}-1} \ldots d\mathbf{x}_1, \tag{10}$$

which is empowered by a reverse diffusion chain of length $T_{\text{trunc}}$, and the encoder (variational posterior) as $q(\mathbf{x}_{T_{\text{trunc}}} \,|\, \mathbf{x}_0)$. Thus we can view $q(\mathbf{x}_{T_{\text{trunc}}})$ defined in (7) as the aggregated posterior (Hoffman & Johnson, 2016; Tomczak & Welling, 2018). In addition to imposing an auto-encoding data-reconstruction loss, the key idea of the AAE (Makhzani et al., 2015) is to also match the aggregated posterior to a fixed prior. This idea differs AAE from a VAE that regularizes the auto-encoder by matching the variational posterior to a fixed prior under the KL divergence. To this end, we introduce a diffusion-based AAE (Diffusion-AAE), whose loss function is defined as

$$\mathcal{L}_{\text{Diffusion-AAE}} = -\mathbb{E}_{q(\mathbf{x}_0)}\mathbb{E}_{q(\mathbf{x}_{T_{\text{trunc}}} \,|\, \mathbf{x}_0)} \log p_\theta(\mathbf{x}_0 \,|\, \mathbf{x}_{T_{\text{trunc}}}) + \mathcal{D}(q(\mathbf{x}_{T_{\text{trunc}}}))||p_\psi(\mathbf{x}_{T_{\text{trunc}}})). \tag{11}$$

Diffusion-AAE has two notable differences from a vanilla AAE: 1) its encoder is fixed and has no learnable parameters, while its prior is not fixed and is optimized to match the aggregated posterior, and 2) its decoder is a reverse diffusion chain, with $T_{\text{trunc}}$ stochastic layers all parameterized by $\theta$.

Note in general as the likelihood in (10) is intractable, the first loss term in (11) is intractable. However, the loss of Diffusion-AAE is upper bounded by the loss of TDPM, as described below.

**Theorem 1.** *The Diffusion-AAE loss in (11) is upper bounded by the TDPM loss in (8):*

$$\mathcal{L}_{\text{Diffusion-AAE}} \leq \mathcal{L}_{\text{TDPM}}.$$

### 3.4 MATCHING THE PRIOR TO AGGREGATED POSTERIOR

Via the loss term $\tilde{\mathcal{L}}_{T_{\text{trunc}}} := \mathcal{D}\left(q(\mathbf{x}_{T_{\text{trunc}}})||p_\psi(\mathbf{x}_{T_{\text{trunc}}})\right)$ in (8), we aim to match the prior $p_\psi(\mathbf{x}_{T_{\text{trunc}}})$ to the aggregated posterior $q(\mathbf{x}_{T_{\text{trunc}}})$ in TDPM. While we have an analytic density function for neither $p$ nor $q$, we can easily draw random samples from both of them. Thus, we explore the use of two different types of statistical distances that can be estimated from samples of both $q$ and $p$. We empirically show that TDPM can achieve good performance regardless of which distance is used for optimization.

One possible statistical distance is based on the idea of GANs (Goodfellow et al., 2014; Arjovsky et al., 2017; Bińkowski et al., 2018), which are widely used to learn implicit distributions from empirical data. In this setting, we use a generator $G_\psi(\cdot) : \mathbb{R}^d \to \mathbb{R}^d$ to transform samples from an isotropic Gaussian $p(\mathbf{z})$ into samples that approximate the corrupted data, and a discriminator $D_\phi(\cdot) : \mathbb{R}^d \to [0, 1]$ to distinguish between the samples from the corrupted data distribution $q(\mathbf{x}_{T_{\text{trunc}}} \,|\, \mathbf{x}_0)$ and the implicit generative distribution $p_\psi(\mathbf{x}_{T_{\text{trunc}}})$. The generator and the discriminator are trained by the following objective $\mathcal{L}_{T_{\text{trunc}}}^{\text{GAN}}$:

$$\min_\psi \max_\phi \quad \mathbb{E}_{\mathbf{x} \sim q(\mathbf{x}_{T_{\text{trunc}}})}[\log D_\phi(\mathbf{x})] + \mathbb{E}_{\mathbf{z} \sim p(\mathbf{z})} \left[\log(1 - D_\phi(G_\psi(\mathbf{z})))\right]. \tag{12}$$

### 3.5 TRAINING ALGORITHM

As the objective in Equation 8 is a sum of different terms, following DDPM (Ho et al., 2020) to fix the terms $\Sigma_\theta(x_t, t) = \sigma_t^2 \boldsymbol{I}$, we can simplify $\frac{1}{T_{\text{trunc}}} \sum_{t=1}^{T_{\text{trunc}}} \mathcal{L}_{t-1}$ as an expectation defined as

$$\mathcal{L}_{\text{simple\_trunc}} = \mathbb{E}_{t,\mathbf{x}_0,\epsilon_t}\left[||\epsilon_t - \epsilon_\theta(\mathbf{x}_t, t)||^2\right], \ \ t \sim \text{Unif}(1, 2, \dots, T_{\text{trunc}}), \ \ \epsilon_t \sim \mathcal{N}(\mathbf{0}, \boldsymbol{I}) \qquad (13)$$

where $\epsilon_t$ is an injected noise at a uniformly sampled timestep index $t$, $\mathbf{x}_t = \sqrt{\bar{\alpha}_t}\mathbf{x}_0 + \sqrt{1 - \bar{\alpha}_t}\epsilon_t$ is a noisy image at time $t$, and $\epsilon_\theta$ is a denoising U-Net that predicts the noise in order to refine the noisy image $\mathbf{x}_t$. Therefore the final simplified version of (8) is constructed as

$$\mathcal{L}_{\text{TDPM}}^{\text{GAN}} = \mathcal{L}_{\text{simple\_trunc}} + \lambda \mathcal{L}_{T_{\text{trunc}}}^{\text{GAN}}, . \qquad (14)$$

While $\lambda$, the weight of $\mathcal{L}_{T_{\text{trunc}}}$, can be tuned, we fix it as one for simplicity. Here the TDPM objective consists of two parts: the denoising part $\epsilon_\theta$ is focused on denoising the truncated chain, getting updated from $\mathcal{L}_{\text{simple\_trunc}}$, while the implicit part $G_\psi$ is focused on minimizing $\mathbb{E}_q[\mathcal{D}\left(q(\mathbf{x}_{T_{\text{trunc}}})||p_\psi(\mathbf{x}_{T_{\text{trunc}}})\right)]$, getting updated from $\mathcal{L}_{T_{\text{trunc}}}^{\text{GAN}}$.

An interesting finding of this paper is that we do not necessarily need to introduce a separate set of parameters $\psi$ for the generator $G_\psi$, as we can simply reuse the same parameters $\theta$ of the reverse diffusion model (*i.e.*, let $\psi = \theta$) without clearly hurting the empirical performance. This suggests that the reverse diffusion process from $T$ to $T_{\text{trunc}}$ could be effectively approximated by a single step using the same network architecture and parameters as the reverse diffusion steps from $T_{\text{trunc}}$ to 0.

Therefore, we provide two configurations to parameterize the implicit distributions. **1)** To save parameters, we let the implicit generator and denoising model share the same U-Net parameters but using different time step indices. Specifically, we first use $\mathbf{x}_{T_{\text{trunc}}} = G_\psi(\mathbf{x}_T) = \epsilon_\theta(\mathbf{x}_T, t{=}T_{\text{trunc}}{+}1)$, where $\mathbf{x}_T \sim \mathcal{N}(\mathbf{0}, \boldsymbol{I})$, to generate a noisy image at time $T_{\text{trunc}}$. **2)** We further explore employing a different model, *e.g.*, StyleGAN2 (Karras et al., 2020a), for the implicit generator, which provides better performance but increases the model size to get $\mathbf{x}_{T_{\text{Trunc}}}$. Then for $t{=}T_{\text{trunc}}, \dots, 1$, we iteratively refine it as $\mathbf{x}_{t-1} = \frac{1}{\sqrt{\alpha_t}}(\mathbf{x}_t - \frac{1-\alpha_t}{\sqrt{1-\bar{\alpha}_t}}\epsilon_\theta(\mathbf{x}_t, t)) + \beta_t\mathbf{z}_t$, where $\mathbf{z}_t \sim N(\mathbf{0}, \boldsymbol{I})$ when $t > 1$ and $\mathbf{z}_1 = \mathbf{0}$. This process is depicted in Algorithms 1 and 2 in the Appendix. For the implementation details, please refer to Appendix D.6 and our code at `https://github.com/JegZheng/truncated-diffusion-probabilistic-models`.

### 3.6 RELATED WORK

In our previous discussions, we have related TDPM to several existing works such as DDPM and AAE. A detailed discussion on other related works is provided in Appendix B.

## 4 EXPERIMENTS

We aim to demonstrate that TDPM can generate good samples faster by using fewer steps of reverse diffusion. We use different image datasets to test our method and follow the same setting as other diffusion models (Ho et al., 2020; Nichol & Dhariwal, 2021; Dhariwal & Nichol, 2021; Rombach et al., 2022) for our backbones. We also have two ways to set up the implicit generator that starts the reverse diffusion. One way is to reuse the denoising network, and the other way is to use a separate network. We try both ways for generating images without any labels. For generating images from text, we use the first way with the LDM backbone. We provide comprehensive details, toy examples, and additional experimental results in Appendices D.4-D.8.

We use FID (lower is better) and Recall (higher is better) to measure the fidelity and diversity, respectively, of the generated images. We use CIFAR-10 (Krizhevsky et al., 2009), LSUN-bedroom, and LSUN-Church (Yu et al., 2015) datasets in unconditional experiments, and CUB-200 (Welinder et al., 2010) and MS-COCO (Lin et al., 2014) for text-to-image experiments. The images consist of $32 \times 32$ pixels for CIFAR-10 and $256 \times 256$ pixels for the other datasets.

### 4.1 EFFICIENCY IN BOTH TRAINING AND SAMPLING

We first look at the results on CIFAR-10. We use DDPM (Ho et al., 2020) or improved DDPM (Nichol & Dhariwal, 2021) as our backbones. We use 4, 49, or 99 steps of reverse diffusion, which correspond

Table 1: Results of unconditional generation on CIFAR-10, with the best FID and Recall in each group marked in bold. To compare TDPM ($T_{\text{Trunc}}$=0) with GAN-based methods, we use DDPM backbone as generator and StyleGAN2 discriminator.

| Method | NFE | FID↓ | Recall↑ |
|---|---|---|---|
| *DDPM backbone* | | | |
| DDPM | 1000 | 3.21 | 0.57 |
| TDPM ($T_{\text{Trunc}}$=99) | 100 | 3.10 | 0.57 |
| TDPM+ ($T_{\text{Trunc}}$=99) | 100 | **2.88** | **0.58** |
| DDIM | 50 | 4.67 | 0.53 |
| TDPM ($T_{\text{Trunc}}$=49) | 50 | 3.30 | 0.57 |
| TDPM+ ($T_{\text{Trunc}}$=49) | 50 | 2.94 | **0.58** |
| TDPM ($T_{\text{Trunc}}$=4) | 5 | 3.34 | 0.57 |
| TDPM+ ($T_{\text{Trunc}}$=4) | 5 | 3.21 | 0.57 |
| *Improved DDPM backbone* | | | |
| Improved DDPM | 4000 | 2.90 | 0.58 |
| TDPM ($T_{\text{Trunc}}$=99) | 100 | 2.97 | 0.57 |
| TDPM+ ($T_{\text{Trunc}}$=99) | 100 | **2.83** | **0.58** |
| Improved DDPM+DDIM | 50 | 3.92 | 0.55 |
| TDPM ($T_{\text{Trunc}}$=49) | 50 | 3.11 | 0.57 |
| TDPM+ ($T_{\text{Trunc}}$=49) | 50 | 2.96 | 0.58 |
| TDPM ($T_{\text{Trunc}}$=4) | 5 | 3.51 | 0.55 |
| TDPM+ ($T_{\text{Trunc}}$=4) | 5 | 3.17 | 0.57 |
| *GAN-based* | | | |
| DDGAN | 4 | 3.75 | 0.57 |
| StyleGAN2 | 1 | 8.32 | 0.41 |
| StyleGAN2-ADA | 1 | **2.92** | **0.49** |
| TDPM ($T_{\text{Trunc}}$=0) | 1 | 7.34 | 0.46 |

Table 2: Results on LSUN-Church and LSUN-Bedroom (resolution $256 \times 256$). Similar to Table 1, TDPM ($T_{\text{Trunc}}$=0) uses DDPM backbone for the generator.

| Method | NFE | Church FID | Bedroom FID |
|---|---|---|---|
| *DDPM backbone* | | | |
| DDPM | 1000 | 7.89 | 4.90 |
| TDPM ($T_{\text{Trunc}}$=99) | 100 | 4.33 | 3.95 |
| TDPM+ ($T_{\text{Trunc}}$=99) | 100 | **3.98** | **3.67** |
| DDIM | 50 | 10.58 | 6.62 |
| TDPM ($T_{\text{Trunc}}$=49) | 50 | 5.35 | 4.10 |
| TDPM+ ($T_{\text{Trunc}}$=49) | 50 | 4.34 | 3.98 |
| TDPM ($T_{\text{Trunc}}$=4) | 5 | 4.98 | 4.16 |
| TDPM+ ($T_{\text{Trunc}}$=4) | 5 | 4.89 | 4.09 |
| *ADM backbone* | | | |
| ADM | 1000 | **3.49** | 1.90 |
| ADM+DDIM | 250 | 6.45 | 2.31 |
| TDPM ($T_{\text{Trunc}}$=99) | 100 | 4.41 | 2.24 |
| TDPM+ ($T_{\text{Trunc}}$=99) | 100 | 3.61 | **1.88** |
| TDPM ($T_{\text{Trunc}}$=49) | 50 | 4.57 | 2.92 |
| TDPM+ ($T_{\text{Trunc}}$=49) | 50 | 3.67 | 1.89 |
| TDPM ($T_{\text{Trunc}}$=4) | 5 | 5.61 | 7.92 |
| TDPM+ ($T_{\text{Trunc}}$=4) | 5 | 4.66 | 4.01 |
| *GAN-based* | | | |
| DDGAN | 4 | 5.25 | - |
| StyleGAN2 | 1 | **3.93** | **3.98** |
| StyleGAN2-ADA | 1 | 4.12 | 7.89 |
| TDPM ($T_{\text{Trunc}}$=0) | 1 | 4.77 | 5.24 |

Table 3: Results of ImageNet-64×64, evaluated with FID and Recall. TDPM+ is built with a pre-trained ADM and an implicit model trained at $T_{\text{Trunc}}$ using StylGAN-XL.

| Method | NFE | FID↓ | Recall↑ |
|---|---|---|---|
| ADM | 1000 | 2.07 | **0.63** |
| TDPM+ ($T_{\text{Trunc}}$=99) | 100 | **1.62** | **0.63** |
| TDPM+ ($T_{\text{Trunc}}$=49) | 50 | 1.77 | 0.58 |
| TDPM+ ($T_{\text{Trunc}}$=4) | 5 | 1.92 | 0.53 |
| StyleGAN-XL (wo PG) | 1 | 3.54 | 0.51 |

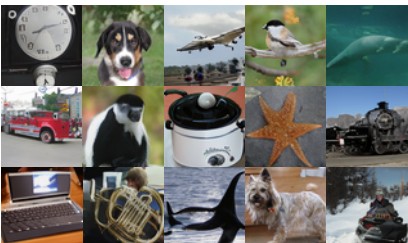

Figure 2: Random generation results of TDPM+ ($T_{\text{Trunc}}$=4) on ImageNet-64×64.

to 5, 50, or 100 number of function evaluations (NFE). For the implicit generator, we either reuse the denoising U-Net or use a StyleGAN2 network (respectively, we call them TDPM and TDPM+). For comparison, we also include DDIM (Song et al., 2020) and DDGAN (Xiao et al., 2022). The comparison with a more diverse set of baselines can be found in Table 9 in Appendix D.7.

Table 1 shows that our TDPM can get good FID with fewer NFE. TDPM+ can get even better FID, and it is the best when NFE=100. Compared with TDPM with 0 steps of reverse diffusion (a GAN with DDPM's U-Net as generator and StyleGAN2 as discriminator) and StyleGAN2, TDPM with more than 0 steps of reverse diffusion has better recall and the FID is as good as StyleGAN2-ADA (a GAN with data augmentation for better training). This means TDPM can largely avoid the mode missing problem in GANs. We show some examples of generated images on CIFAR-10 in Figure 13.

We also check how fast TDPM can train and sample. In training, we count how many images TDPM needs to well fit the truncated diffusion chain and the implicit prior. Figure 3 shows that when we use fewer steps of reverse diffusion, the diffusion part needs less time to train. But the implicit prior needs more time to train because it has to model a harder distribution, *e.g.*, fitting the implicit prior with 4 diffusion steps needs similar time to directly fit it on the data. When we use 99 steps of reverse diffusion, the diffusion chain and the implicit prior need similar time to train, and the whole model trains faster than both GAN and DDPM. In sampling, we compare TDPM with 0, 1, 4, 49, or 99 steps of reverse diffusion. We report both FID and the sampling time (s/image) on one NVIDIA V100 GPU in Figure 4. When we use 4 steps of reverse diffusion, the FID is much lower than 0 steps, and the sampling time is slightly longer. When we use more steps of reverse diffusion, the FID goes down

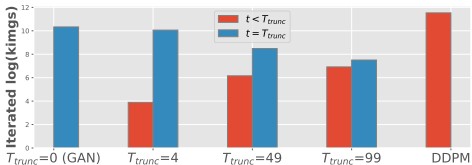

Figure 3: The required iterations (measured with iterated images) to converge in the training. The iterations for $t < T_{\text{Trunc}}$ ($\epsilon_\theta$) and $t = T_{\text{Trunc}}$ ($G_\psi$) are marked in red and blue, respectively.

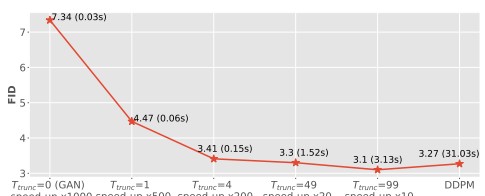

Figure 4: Evolution of FID and corresponding GPU time (s/image) across different timesteps in the sampling stage.

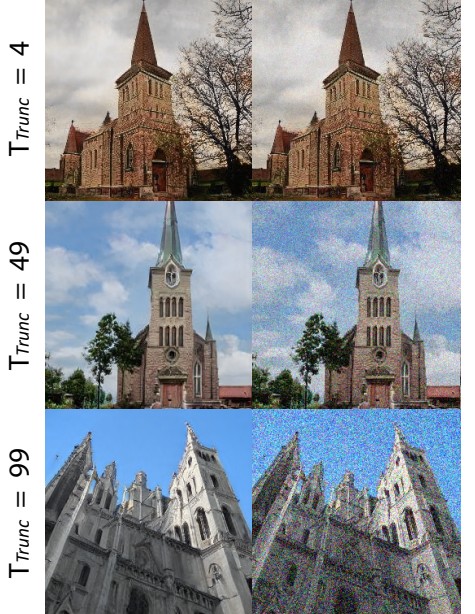

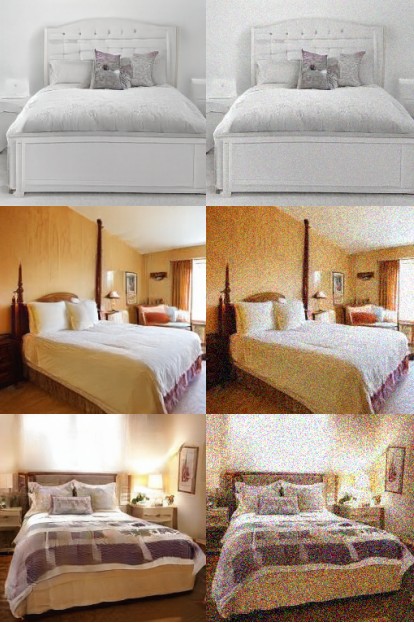

Figure 5: Randomly generated images of TDPM using ADM (Dhariwal & Nichol, 2021) backbone on LSUN-Church and LSUN-Bedroom ($256 \times 256$), with $T_{\text{trunc}} = 4, 49$, and 99. Note NFE $= T_{\text{trunc}} + 1$ in TDPM. Each group presents generated samples from the full model $p_\theta(\mathbf{x}_0)$ (left) and its implicit prior $\sim p_\theta(\mathbf{x}_{T_{\text{trunc}}})$ (right); the full model sample is obtained by refining the implicit prior sample via the truncated reverse diffusion chain.

slowly, but the sampling time goes up linearly. When we use 99 steps of reverse diffusion, the FID of TDPM is better than DDPM with 1000 steps. Because the FID does not change much when we use more steps of reverse diffusion, we suggest using a small number of steps, such as 4 or more, to balance the quality and speed of generation.

## 4.2 RESULTS ON HIGHER-RESOLUTION AND MORE DIVERSE IMAGE DATASETS

To test the performance of the proposed truncation method on high-resolution images, we train TDPM using two different diffusion models, DDPM (Ho et al., 2020) and ADM (Dhariwal & Nichol, 2021), as backbones on two datasets of $256 \times 256$ resolution, LSUN-Church and LSUN-Bedroom (Yu et al., 2015). We compare the FIDs of TDPM with those of the backbone models and some state-of-the-art GANs in Tables 2. The results show that TDPM can generate images of similar quality with much smaller truncation steps $T_{\text{trunc}}$, which means that it can produce images significantly faster than the backbone models. We also visualize the samples from the implicit distribution $\mathbf{x}_{T_{\text{trunc}}} \sim p_\theta(\mathbf{x}_{T_{\text{trunc}}})$ that TDPM generates and the corresponding $\mathbf{x}_0$ that it finishes at the end of reverse chain in Figure 5.

We further evaluate TDPM on ImageNet-1K (with resolution $64 \times 64$) that exhibits high diversity. Here we adopt the TDPM+ configuration, where we use a pre-trained ADM (Dhariwal & Nichol, 2021) checkpoint for $t < T_{\text{Trunc}}$ and train a StyleGAN-XL (Sauer et al., 2022) based implicit model at $t = T_{\text{Trunc}}$ (for simplicity, we choose to not use the progressive growing pipeline of StyleGAN-XL; See Appendix D.6 for more details). We compare both FID and Recall with our backbone models in Table 3 and show example generations in Figure 2. Similar to our observations in Table 1, TDPM has good generation quality with small truncation steps $T_{\text{trunc}}$. Moreover, properly training an implicit model at $T_{\text{trunc}}$ can further improve the performance of the backbone.

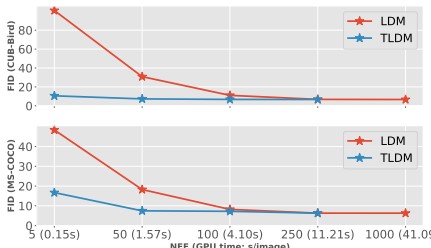

Figure 6: Quantitative text-to-image results (FID and GPU time) across different NFE.

Table 4: Numerical results of Figure 6. The GPU time of sampling (s/image) is measured on one NVIDIA A100.

| | | CUB-Bird | | MS-COCO | |
|---|---|---|---|---|---|
| NFE | GPU time | LDM | TLDM | LDM | TLDM |
| 5 | 0.15 | 100.81 | 10.59 | 48.41 | 16.7 |
| 50 | 1.57 | 30.85 | 7.32 | 18.25 | 7.47 |
| 100 | 4.10 | 11.07 | 6.79 | 8.2 | 7.22 |
| 250 | 11.21 | 6.82 | 6.72 | 6.3 | 6.29 |
| 1000 | 41.09 | 6.68 | - | 6.29 | - |

NFE=100 (T$_{Trunc}$ = 99)   NFE=50 (T$_{Trunc}$ = 49)   NFE=5 (T$_{Trunc}$ = 4)

A bird with brown wings, black back, and red head.

A green train is coming down the tracks.

Figure 7: Example text-to-image generation results of LDM and TLDM (*i.e.*, TDPM with LDM backbone) finetuned on CUB-200 (top row) or MS-COCO (bottom row), setting the number of times iterating through the reverse diffusion U-Net as 100 (left column), 50 (middle column), or 5 (right column).

### 4.3 TEXT-TO-IMAGE GENERATION

Besides unconditional generation tasks, we develop for text-to-image generation the TLDM, a conditional version of TDPM that leverages as the backbone the LDM of Rombach et al. (2022), which is a state-of-the-art publicly released model with 1.45B parameters pre-trained on LAION-400M (Schuhmann et al., 2021). LDM consists of a fixed auto-encoder for pixel generation and a latent-diffusion module to connect text and image embeddings. Here we fine-tune its latent-diffusion part on CUB-200 and MS-COCO datasets with 25K and 100K steps as the baseline. Similar to the unconditional case, we fine-tune with the LDM loss for $t < T_{\text{Trunc}}$ and GAN loss for $t = T_{\text{Trunc}}$. More details about the setting can be found in Appendix D.6.

The results of LDM with different DDIM sampling steps and TLDM with different truncated steps are summarized in Figure 6 and Table 4. Similar to applying diffusion directly on the original image-pixel space, when the diffusion chain is applied in the latent space, we observe TLDM can achieve comparable or better performance than LDM even though it has shortened the diffusion chain of LDM to have much fewer reverse diffusion steps. For the case that NFE is as small as 5, we note although the FID of TLDM has become higher due to using fewer diffusion steps, the generated image using TLDM at NFE=5 is still visually appealing, as shown in Figure 7. Compared with 50 and 250 steps using LDM, the sampling speed of TLDM using 5 steps is 10 and 50 times faster, respectively, while largely preserving generation quality. We provide additional text-to-image generation results of TLDM in Appendix D.8.

## 5 CONCLUSION

In this paper, we investigate how to reduce the trajectory length of the diffusion chain to achieve efficient sampling without loss of generation quality. We propose truncated diffusion probabilistic modeling (TDPM) that truncates the length of a diffusion chain. In this way, TDPM can use a much shorter diffusion chain, while being required to start the reverse denoising process from an intractable distribution. We propose to learn such a distribution with an implicit generative model powered by the same U-Net used for denoising diffusion, and validate with multiple ways to learn the implicit distribution to ensure the robustness of the proposed TDPM. We reveal that TDPM can be cast as an adversarial auto-encoder with a learnable implicit prior. We conduct extensive experiments on both synthetic and real image data to demonstrate the effectiveness of TDPM in terms of both sample quality and efficiency, where the diffusion chain can be shortened to have only a few steps.

ACKNOWLEDGMENTS

H. Zheng and M. Zhou acknowledge the support of NSF-IIS 2212418 and IFML.

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

# A   PROOF

*Proof of Theorem 1.* As the last terms in both losses are the same, we only need to show that the first term in (11) is smaller than or equal to $\mathcal{L}_0 + \sum_{t=2}^{T_{\text{trunc}}} \mathcal{L}_{t-1}$ in (8). Using Jensen's inequality, we have

$$
\begin{aligned}
&- \mathbb{E}_{q(\mathbf{x}_0)} \mathbb{E}_{q(\mathbf{x}_{T_{\text{trunc}}} \mid \mathbf{x}_0)} \log p_\theta(\mathbf{x}_0 \mid \mathbf{x}_{T_{\text{trunc}}}) \\
&= -\mathbb{E}_{q(\mathbf{x}_0)} \mathbb{E}_{q(\mathbf{x}_{T_{\text{trunc}}} \mid \mathbf{x}_0)} \log \mathbb{E}_{q(\mathbf{x}_{1:T_{\text{trunc}}-1} \mid \mathbf{x}_0, \mathbf{x}_{T_{\text{trunc}}})} \left[ \frac{p(\mathbf{x}_{0:T_{\text{trunc}}-1} \mid \mathbf{x}_{T_{\text{trunc}}})}{q(\mathbf{x}_{1:T_{\text{trunc}}-1} \mid \mathbf{x}_0, \mathbf{x}_{T_{\text{trunc}}})} \right] \\
&\leq -\mathbb{E}_{q(\mathbf{x}_0)} \mathbb{E}_{q(\mathbf{x}_{T_{\text{trunc}}} \mid \mathbf{x}_0)} \mathbb{E}_{q(\mathbf{x}_{1:T_{\text{trunc}}-1} \mid \mathbf{x}_0, \mathbf{x}_{T_{\text{trunc}}})} \log \frac{p(\mathbf{x}_{0:T_{\text{trunc}}-1} \mid \mathbf{x}_{T_{\text{trunc}}})}{q(\mathbf{x}_{1:T_{\text{trunc}}-1} \mid \mathbf{x}_0, \mathbf{x}_{T_{\text{trunc}}})} \\
&= -\mathbb{E}_{q(\mathbf{x}_0)} \mathbb{E}_{q(\mathbf{x}_{1:T_{\text{trunc}}} \mid \mathbf{x}_0)} \log \left[ \frac{p(\mathbf{x}_{0:T_{\text{trunc}}-1})}{q(\mathbf{x}_{1:T_{\text{trunc}}} \mid \mathbf{x}_0)} \frac{q(\mathbf{x}_{T_{\text{trunc}}} \mid \mathbf{x}_0)}{p(\mathbf{x}_{T_{\text{trunc}}})} \right] \\
&= \left( -\mathbb{E}_{q(\mathbf{x}_0)} \mathbb{E}_{q(\mathbf{x}_{1:T_{\text{trunc}}} \mid \mathbf{x}_0)} \log \frac{p(\mathbf{x}_{0:T_{\text{trunc}}-1})}{q(\mathbf{x}_{1:T_{\text{trunc}}} \mid \mathbf{x}_0)} \right) - \mathbb{E}_{q(\mathbf{x}_0)} \mathbb{E}_{q(\mathbf{x}_{T_{\text{trunc}}} \mid \mathbf{x}_0)} \log \frac{q(\mathbf{x}_{T_{\text{trunc}}} \mid \mathbf{x}_0)}{p(\mathbf{x}_{T_{\text{trunc}}})} \\
&= \left( \sum_{t=1}^{T_{\text{trunc}}} \mathcal{L}_{t-1} + \mathcal{L}_{T_{\text{trunc}}} \right) - \mathcal{L}_{T_{\text{trunc}}} \\
&= \sum_{t=1}^{T_{\text{trunc}}} \mathcal{L}_{t-1},
\end{aligned}
\tag{15}
$$

where the second to last equality follows the same derivation of the ELBO in Ho et al. (2020).   □

# B   RELATED WORK

Diffusion probabilistic models (Sohl-Dickstein et al., 2015; Ho et al., 2020) employ a forward Markov chain to diffuse the data to noise and learn the reversal of such a diffusion process. With the idea of exploiting the Markov operations (Goyal et al., 2017; Alain et al., 2016; Bordes et al., 2017), diffusion models achieve great success and inspire a variety of tasks including image generation and audio generation (Kong et al., 2020; Chen et al., 2020; Jolicoeur-Martineau et al., 2020; Vahdat et al., 2021). Recently, plenty of studies have been proposed to generalize diffusion model to continuous time diffusion and improve the diffusion models in likelihood estimation (Vincent, 2011; Song & Ermon, 2020; 2019; Nichol & Dhariwal, 2021; Song et al., 2021b;a; Kingma et al., 2021).

Another mainstream is to improve the sampling efficiency of diffusion models, which are known for their enormous number of sampling steps. Luhman & Luhman (2021) improve diffusion processes with knowledge distillation and San-Roman et al. (2021) propose a learnable adaptive noise schedule. Song et al. (2020) and Kong & Ping (2021) exploit non-Markovian diffusion processes and shorten the denoising segments. Jolicoeur-Martineau et al. (2021) and Huang et al. (2021) use better SDE solvers for continuous-time models. Aside from these works, recently other types of generative models such as VAEs (Kingma & Welling, 2013), GANs (Goodfellow et al., 2014), and autoregressive models (van den Oord et al., 2016) have been incorporated to diffusion models. They are shown to benefit each other (Xiao et al., 2022; Pandey et al., 2022; Meng et al., 2021) and have a closer relation to our work. Xiao et al. (2022) consider the use of implicit models (Huszár, 2017; Mohamed & Lakshminarayanan, 2016; Tran et al., 2017; Yin & Zhou, 2018; Li & Malik, 2018) to boost the efficiency of diffusion models, where they deploy implicit models in each denoising step, which has higher difficulty in the training as the number of diffusion steps increases. Pandey et al. (2022) build diffusion models on top of the output of VAEs for refinement. Our work is also related if viewing TDPM as a diffusion model on top of an implicit model, where the implicit model can be parameterized with the U-Net or a separate network.

# C   DISCUSSION

**Potential societal impacts**: This paper proposes truncated diffusion probabilistic model as a novel type of diffusion-based generative model. The truncated part can be trained as implicit generative models such as GANs jointly or independently with the diffusion part. The capacities of truncated diffusion probabilistic models are competitive to existing diffusion-based ones and efficiency is largely improved. On the contrary of these positive effects, some negative perspectives could also be

seen, depending on how the models are used. One major concern is the truncated diffusion technique proposed in this paper could potentially be a way to hack the existing diffusion models if the implicit models are maliciously used to fit the intermediate steps. For example, for some existing diffusion models, for safety concerns, the model's capacity to generate private data needs to be locked by hiding the diffusion ending point into an unknown distribution. The technique of TDPM could be used to crack these existing online diffusion models by providing intermediate noisy images or fine-tuning the first few steps with TDPM to unlock the capacity. Besides, the capacity of generating good images can also be misused to generate ill-intentioned images at a much lower cost.

**Discussions**: In this work, we mainly focus on reducing the length of the diffusion chain of a finite-time diffusion model. Our model has shown its effectiveness in improving finite-time diffusion models and it is non-trivial to further explore our model on continuous-time diffusion models (Song et al., 2021b). Moreover, while in this paper DDPM is the primary baseline, TDPM can also be built on other recent diffusion models. While $p_\theta(\mathbf{x}_{T_{\text{trunc}}})$ is parameterized as an implicit distribution, it can also be formulated as a semi-implicit distribution (Yin & Zhou, 2018), which allows it to be approximated with a Gaussian generator. Xiao et al. (2022) also present a closely related work. While we share the same spirit to reduce the length of the diffusion chain, these two strategies are not conflicting with each other. In future work we will look into the integration of these different strategies. There also exists plenty of options in approximating $p_\theta(\mathbf{x}_{T_{\text{trunc}}})$. When truncating the diffusion chain to be short, the implicit distribution still faces multi-modal and needs to fit with different methods depending upon the properties that we need. For example, in order to capture all modes, a VAE would be preferred, like done in Pandey et al. (2022). Below we provide an alternative method proposed in Zheng & Zhou (2021) to fit the truncated distribution. Besides the training, it's also an open question whether TDPM can be incorporated into more advanced architectures to have further improvements and we leave this exploration for future work.

# D ALGORITHM DETAILS AND COMPLEMENTARY RESULTS

Below we provide additional algorithm details and complementary experimental results.

## D.1 ADDITIONAL ANALYSIS ON THE PARAMETERIZATION OF THE IMPLICIT GENERATOR

As shown in Section 3, in general, the objective of TDPM consists of the training of the diffusion model $\epsilon_\theta$ (a U-Net architecture (Ronneberger et al., 2015)) with simple loss of DDPM $\mathcal{L}_{\text{simple}}$ and the training of an implicit prior model $G_\psi$ with objective $\mathcal{L}_{T_{\text{trunc}}}^{\text{GAN}}$. Without loss of generality, in our main paper, we show two configurations to parameterize the implicit part for $t = T_{\text{trunc}}$: 1) the implicit generator shares the same U-Net architecture used for $0 < t < T_{\text{trunc}}$; 2) the implicit generator is instantiated with a separate network. Below we explain this two configurations (denoted as TDPM+ in the main paper).

**Configuration 1)**: At $t = T_{\text{trunc}}$, the Unet generates the noisy image at the truncated step: $\mathbf{x}_{T_{\text{trunc}}} = \epsilon_\theta(\mathbf{x}_{T_{\text{trunc}}+1}, t = T_{\text{trunc}} + 1)$, where $\mathbf{x}_{T_{\text{trunc}}+1} \sim \mathcal{N}(0, I)$ is the pure noise image whose pixels are *iid* sampled from standard normal. For $t = T_{\text{trunc}}, T_{\text{trunc}} - 1, \ldots, 1$, the same Unet iteratively refines the noisy images by letting $\mathbf{x}_{t-1} = \frac{1}{\sqrt{\bar{\alpha}_t}}(\mathbf{x}_t - \frac{1-\alpha_t}{\sqrt{1-\bar{\alpha}_t}}\epsilon_{t-1}) + \beta_t \mathbf{z}_t$; $\mathbf{z}_{t>1} \sim \mathcal{N}(0, I), \mathbf{z}_1 = \mathbf{0}$, where $\epsilon_{t-1} = \epsilon_\theta(\mathbf{x}_t, t)$ is the predicted noise by the Unet.

Under this setting, the Unet-based generator plays two roles at the same time and the training will be more challenging than using two different generators here. However, we can also see as $T_{\text{trunc}}$ gets larger, the distribution of $p(\mathbf{x}_{T_{\text{trunc}}})$ will become more similar to a noise distribution, and generating the noisy images will be more like generating noises. In this case, being able to generate both noisy images and predicting noise becomes easier for the generator.

**Configuration 2) (TDPM+)**: Unlike previous configuration, where the implicit generator at step $t = T$ shares the same U-Net architecture with $t < T_{\text{trunc}}$. Another way is to parameterize $G_\psi$ with a separate generator. Although this configuration increases the total parameter of the generative model, it allows the model has better flexibility in the training stage. For example, these two networks can be trained in parallel or leverage a pre-trained model. In our paper, we conduct the experiments by using Stylegan2 generator architecture Karras et al. (2020b) for $t = T_{\text{trunc}}$, resulting in an increase of 19M and 28M for the generator parameters when handling $32 \times 32$ and $256 \times 256$ images.

The process of training and sampling of these configurations are summarized in Algorithm 1 and 2.

---

**Algorithm 1** Training

1: **repeat**
2:    $\mathbf{x}_0 \sim q(\mathbf{x}_0)$
3:    $t \sim \text{Uniform}(\{1, \ldots, T_{\text{trunc}}\})$
4:    $\epsilon_t \sim \mathcal{N}(\mathbf{0}, \boldsymbol{I}), \mathbf{z} \sim \mathcal{N}(\mathbf{0}, \boldsymbol{I})$
5:    Update with (14)
6: **until** converged

---

**Algorithm 2** Sampling

1: $\mathbf{x}_{T_{\text{trunc}}+1} \sim \mathcal{N}(\mathbf{0}, \boldsymbol{I})$
2: **if** $G_\psi$ shared with $\epsilon_\theta$ **then**
3:    $\mathbf{x}_{T_{\text{trunc}}} = \epsilon_\theta(\mathbf{x}_{T_{\text{trunc}}+1}, T_{\text{trunc}} + 1)$
4: **else**
5:    $\mathbf{x}_{T_{\text{trunc}}} = G_\psi(\mathbf{x}_{T_{\text{trunc}}+1})$
6: **end if**
7: **for** $t = T_{\text{trunc}}, \ldots, 1$ **do**
8:    $\mathbf{z}_t \sim \mathcal{N}(\mathbf{0}, \boldsymbol{I})$ if $t > 1$, else $\mathbf{z}_1 = \mathbf{0}$
9:    $\mathbf{x}_{t-1} = \frac{1}{\sqrt{\alpha_t}} \left( \mathbf{x}_t - \frac{1-\alpha_t}{\sqrt{1-\bar{\alpha}_t}} \epsilon_\theta(\mathbf{x}_t, t) \right) + \beta_t \mathbf{z}_t$
10: **end for**
11: **return** $\mathbf{x}_0$

---

## D.2 ALTERNATIVES OF LEARNING THE IMPLICIT DISTRIBUTION

Another possible statistical distance is based on conditional transport (Zheng & Zhou, 2021), which is proposed to balance the model-seeking and mode-covering behaviors when fitting an empirical data distribution. In this setting, we use the same generator $G_\psi$ as before, but instead of a discriminator, we use a conditional distribution $\pi_\eta$ parameterized by $\eta$ to find an optimized mapping between the samples of $p$ and $q$, and a critic $\phi$ to measure the point-to-point cost $c_\phi$ in the feature space. The generator, the conditional distribution, and the critic are trained by the following objective $\mathcal{L}_{T_{\text{trunc}}}^{\text{CT}}$:

$$\min_{\psi, \eta} \max_{\phi} \mathbb{E}_{\mathbf{x} \sim q(\mathbf{x}_{T_{\text{trunc}}})} \left[ \mathbb{E}_{G_\psi(\mathbf{z}) \sim \pi_\eta(G_\psi(\mathbf{z}) \mid \mathbf{x}_{T_{\text{trunc}}})} c_\phi(\mathbf{x}_{T_{\text{trunc}}}, G_\psi(\mathbf{z})) \right]$$
$$+ \mathbb{E}_{\mathbf{z} \sim p(\mathbf{z})} \left[ \mathbb{E}_{\mathbf{x}_{T_{\text{trunc}}} \sim \pi_\eta(\mathbf{x}_{T_{\text{trunc}}} \mid G_\psi(\mathbf{z}))} c_\phi(\mathbf{x}_{T_{\text{trunc}}}, G_\psi(\mathbf{z})) \right]. \tag{16}$$

Similar to (14), we fit TDPM-CT with following loss

$$\mathcal{L}_{\text{TDPM}}^{\text{CT}} = \mathcal{L}_{\text{simple\_trunc}} + \lambda \mathcal{L}_{T_{\text{trunc}}}^{\text{CT}}. \tag{17}$$

We empirically find out this objective has no significant difference than using GAN objective shown in Equation 14 in performance-wise as long as the generator is well trained.

## D.3 CONDITIONAL TRUNCATED DIFFUSION PROBABILISTIC MODELS

For conditional generation, we extend (14) and derive a conditional version of TDPM:

$$\mathcal{L}_{\text{TDPM}}^{\mathbf{c}} = \mathcal{L}_{\text{simple\_trunc}}^{\mathbf{c}} + \lambda \mathcal{L}_{T_{\text{trunc}}}^{\mathbf{c}}, \tag{18}$$

where $\mathcal{L}_{\text{simple\_trunc}}^{\mathbf{c}}$ aims to train the conditional diffusion model with

$$\mathcal{L}_{\text{simple\_trunc}}^{\mathbf{c}} = \mathbb{E}_{\mathbf{c}} \mathbb{E}_{t, \mathbf{x}_0 \mid \mathbf{c}, \epsilon_t} \left[ ||\epsilon_t - \epsilon_\theta(\mathbf{x}_t, \mathbf{c}, t)||^2 \right], \ \ t \sim \text{Unif}(1, 2, \ldots, T_{\text{trunc}}), \ \ \epsilon_t \sim \mathcal{N}(\mathbf{0}, \boldsymbol{I}), \tag{19}$$

and the truncated distribution $\mathcal{L}_{T_{\text{trunc}}}^{\mathbf{c}}$ can be fitted with GAN or CT:

$$\min_{\psi} \max_{\phi} \ \ \mathbb{E}_{\mathbf{c}} \left[ \mathbb{E}_{\mathbf{x} \sim q(\mathbf{x}_{T_{\text{trunc}}} \mid \mathbf{c})} [\log D_\phi(\mathbf{x} \mid \mathbf{c})] + \mathbb{E}_{\mathbf{z} \sim p(\mathbf{z})} \left[ \log(1 - D_\phi(G_\psi(\mathbf{z}, \mathbf{c)) \mid \mathbf{c})] \right]. \tag{20}$$

$$\min_{\psi, \eta} \max_{\phi} \mathbb{E}_{\mathbf{c}} \left[ \mathbb{E}_{\mathbf{x} \sim q(\mathbf{x}_{T_{\text{trunc}}} \mid \mathbf{c})} \left[ \mathbb{E}_{G_\psi(\mathbf{z}) \sim \pi_\eta(G_\psi(\mathbf{z}, \mathbf{c}) \mid \mathbf{x}_{T_{\text{trunc}}}, \mathbf{c})} c_\phi(\mathbf{x}_{T_{\text{trunc}}}, G_\psi(\mathbf{z}, \mathbf{c})) \right] \right.$$
$$\left. + \mathbb{E}_{\mathbf{z} \sim p(\mathbf{z})} \left[ \mathbb{E}_{\mathbf{x}_{T_{\text{trunc}}} \sim \pi_\eta(\mathbf{x}_{T_{\text{trunc}}} \mid G_\psi(\mathbf{z}, \mathbf{c}), \mathbf{c})} c_\phi(\mathbf{x}_{T_{\text{trunc}}}, G_\psi(\mathbf{z}, \mathbf{c})) \right] \right]. \tag{21}$$

## D.4 ANALYSIS ON TOY EXPERIMENTS

Although we present image experiments in the main paper, our studies were firstly justified our method on synthetic toy data as a proof of concept. We adopt representative 2D synthetic datasets used in prior works (Gulrajani et al., 2017; Zheng & Zhou, 2021), including Swiss Roll, Double Moons, 8-modal, and 25-modal Gaussian mixtures with equal component weights. We use an empirical

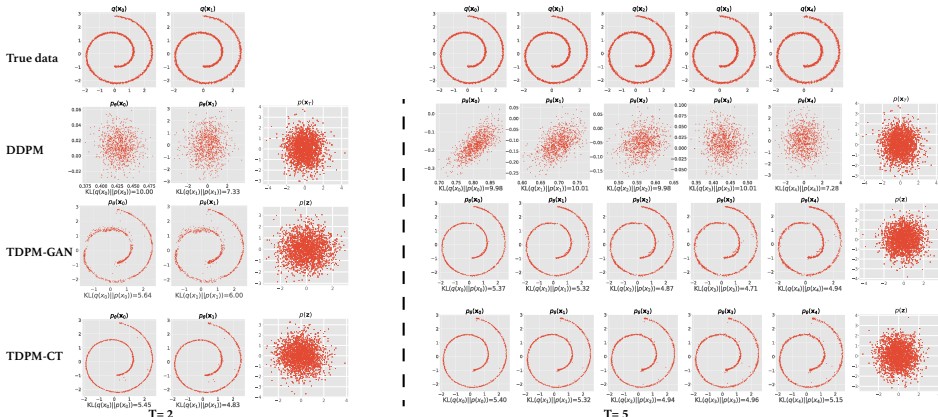

Figure 8: A comparison of DDPM (Ho et al., 2020), TDPM-GAN, and TDPM-CT on Swiss Roll toy data. We show the effects of a truncated diffusion chain with length $T=2$ and $T=5$ ($T_{\text{Trunc}}=1$ and $T_{\text{Trunc}}=4$). The first row displays the true distribution from $q(\mathbf{x}_0)$ to $q(\mathbf{x}_{T-1})$. Each row below the first one represents the corresponding denoising distribution $p_\theta(\mathbf{x}_{t-1}\,|\,\mathbf{x}_t)$. DDPM assumes $p(\mathbf{x}_T)=\mathcal{N}(\mathbf{0},\boldsymbol{I})$ and we can observe a gap between the true data distribution $q(\mathbf{x}_{T-1})$ and its generative distribution $p_\theta(\mathbf{x}_{T-1})$. TDPM learns $p_\theta(\mathbf{x}_{T_{\text{trunc}}})$, which can be observed to well approximate the true $q(\mathbf{x}_{T-1})$, which helps the model successfully recover the clean data distribution $q(\mathbf{x}_0)$. Below each model, we report empirical KL divergence between data and generative distributions as the quantitative metric. More results on different toy data can be found in Appendix D.

sample set $\mathcal{X}$, consisting of $|\mathcal{X}|=2,000$ samples and illustrate the generated samples after 5000 training epochs. We take 20 grids in the range $[-10, 10]$ for both the $x$ and $y$ axes to approximate the empirical distribution of $\hat{p}_\theta$ and $\hat{q}$, and report the corresponding forward KL $D_{\text{KL}}(\hat{q}||\hat{p}_\theta)$ as the quantitative evaluation metric.

Figure 8 shows the results on the Swiss Roll data. We present a short chain with $T=2$ and a longer chain with $T=5$ to show the impacts of the number of diffusion steps. The first row shows that the data distribution is diffused with accumulated noise, and with more steps the diffused distribution will be closer to an isotropic Gaussian distribution. As one can see, truncating the diffusion chain to a short length will result in a clear gap between $q(\mathbf{x}_{T_{\text{trunc}}})$ and $\mathcal{N}(\mathbf{0},\boldsymbol{I})$. When DDPM (shown in the second row) samples from the isotropic Gaussian distribution, it becomes hard to recover the original data distribution from pure noise with only a few steps. Although we can see DDPM can get slightly improved with a few more steps ($T=5$), as long as $q(\mathbf{x}_T)$ is not close to Gaussian, DDPM can hardly recover the data distribution. By contrast, as shown in the third and fourth rows, TDPM successfully approximates the non-Gaussian $q(\mathbf{x}_{T_{\text{trunc}}})$ with its implicit generator, and we can see the remaining part of the truncated chain is gradually recovered by the denoising steps. From both visualizations and $D_{\text{KL}}(\hat{q}||\hat{p}_\theta)$, we can see that TDPM is able to fit every step in such short chains.

TDPM-GAN and TDPM-CT both succeed in fitting $p_\theta(\mathbf{x}_{T_{\text{trunc}}})$ but the latter one fits slightly better when the diffusion length is 2. When the length increases to 5, fitting the implicit distribution with GAN becomes easier. This observation demonstrate a benefit of combining the diffusion models and GANs. If the implicit generator is sufficiently powerful to model $q(\mathbf{x}_{T_{\text{trunc}}})$, then the number of steps in need can be compressed to a small number. On the contrary, if the implicit generator cannot capture the distribution, we need more steps to facilitate the fitting of the data distribution.

Shown in Figure 9-Figure 11, we can see 8-modal Gaussian is more similar to an isotropic Gaussian after getting diffused, thus DDPM can recover a distribution similar to data with 5 steps. On 25-Gaussians, we can observe GAN does not suffer from mode-collapse and provide a better approximation than CT, which results in better data distribution recovery in the final step.

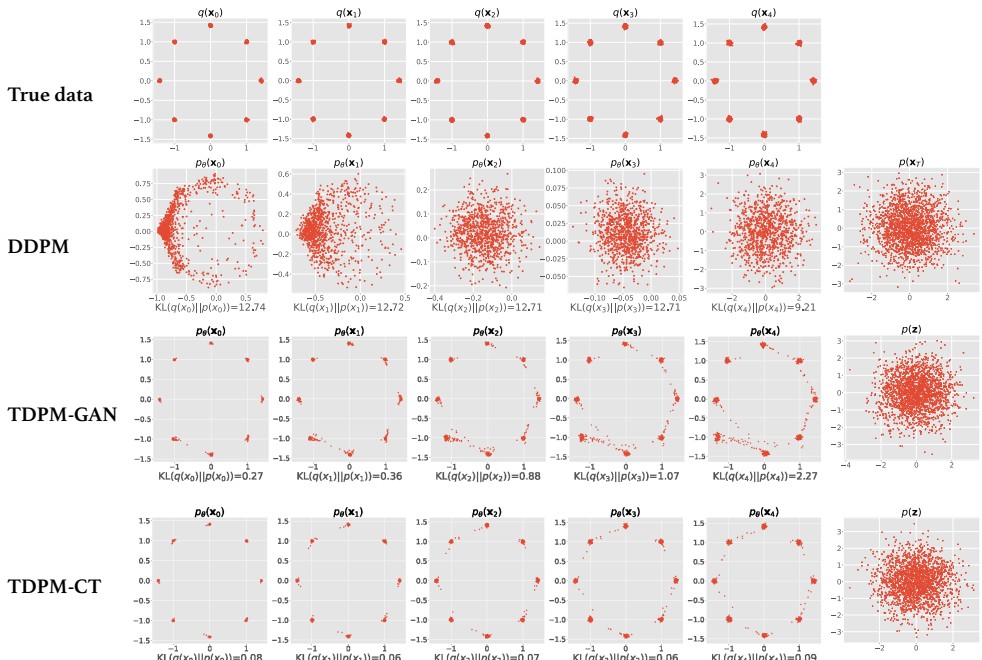

Figure 9: Analogous results to Figure 8 using 8-modal Gaussian data.

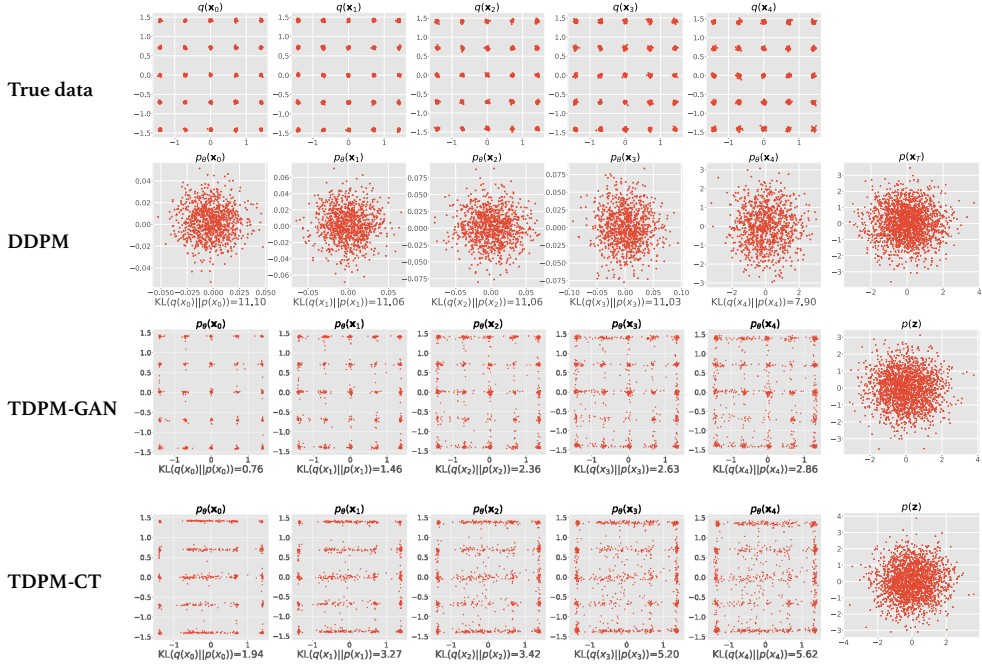

Figure 10: Analogous results to Figure 8 using 25-modal Gaussian data.

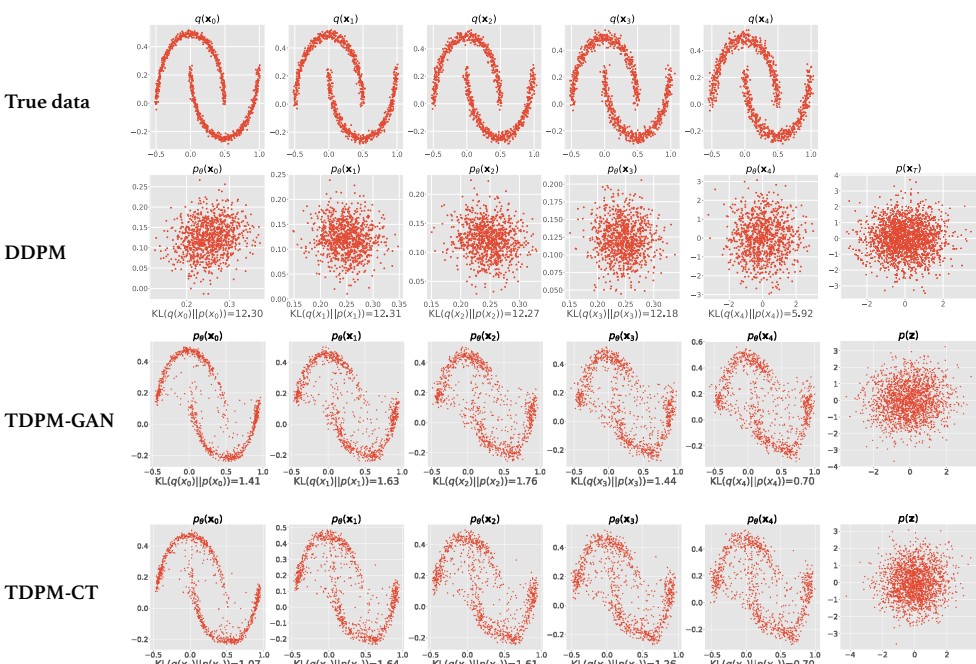

Figure 11: Analogous results to Figure 8 using Double Moons data.

## D.5 ADDITIONAL ABLATION STUDIES

**Using Pre-trained diffusion backbones:** Different from the default setting, here we put the implicit model of TDPM+ trained at $t = T_{\text{trunc}}$ and a pre-trained DDPM model[1] in the same pipeline of sampling. In this case we do not need to spend any time on pretraining the DDPM model, and only need to train the implicit model for $t = T_{\text{trunc}}$. As shown in Table 5, when combined with a pre-trained DDPM for $t < T_{\text{trunc}}$, the generation performance of TDPM trained under this two-step procedure is comparable to TDPM trained end-to-end.

Table 5: Results of adding a separately trained implicit generator to a pre-trained diffusion model on CIFAR-10.

| Model ($t = T_{\text{Trunc}}$) | Model ($t < T_{\text{Trunc}}$) | FID↓ |
|---|---|---|
| TDPM+ ($T_{\text{trunc}}$=99) | TDPM+ (DDPM backbone) | 2.88 |
| | pre-trained DDPM | 2.85 |
| | TDPM+ (improved-DDPM backbone) | 2.83 |
| | pre-trained improved-DDPM | 2.25 |
| TDPM+ ($T_{\text{trunc}}$=49) | TDPM+ | 2.94 |
| | pre-trained DDPM | 3.05 |
| | TDPM+ (improved-DDPM backbone) | 2.96 |
| | pre-trained improved-DDPM | 2.60 |
| TDPM+ ($T_{\text{trunc}}$=4) | TDPM+ (DDPM backbone) | 3.21 |
| | pre-trained DDPM | 3.25 |
| | TDPM+ (improved-DDPM backbone) | 3.17 |
| | pre-trained improved-DDPM | 2.95 |

**Sensitivity to noise schedule**: Nichol & Dhariwal (2021) show the noise schedule affects the training of DDPM. Here we examine if TDPM is sensitive to the choice of noise schedule. We compare the linear schedule with cosine schedule, which adds noise in a milder manner. The results on CIFAR-10 are reported in Table 6, which suggest that TDPM is not sensitive to the choice between these two schedules.

Table 6: Ablation study with different noise schedules on CIFAR-10. The number before and after "/" denotes the FID using linear and cosine schedules, respectively.

| Model | Steps | FID↓ (linear / cosine) |
|---|---|---|
| TDPM-GAN | $T_{\text{trunc}}$=99 | 3.10 / 3.47 |
| TDPM-GAN | $T_{\text{trunc}}$=49 | 3.30 / 3.16 |
| TDPM-CT | $T_{\text{trunc}}$=99 | 3.69 / 3.62 |
| TDPM-CT | $T_{\text{trunc}}$=49 | 3.97 / 3.24 |

**On the choice of truncated step**: As the diffused distribution could facilitate the learning of the implicit generator $G_{\psi}$ (Arjovsky & Bottou, 2017), where we could observe by increasing the number of diffusion steps, the FID of TDPM consistently gets better. A natural question is on which step should we truncate the diffusion chain. We study the signal-to-noise ratio (SNR) of different diffusion step. Based on $q(\mathbf{x}_t|\mathbf{x}_0) = \mathcal{N}(\sqrt{\bar{\alpha}_t}\mathbf{x}_0, 1 - \bar{\alpha}_t I)$, we calculate SNR as

$$\text{SNR} = \frac{\sqrt{\bar{\alpha}_t}}{\sqrt{1 - \bar{\alpha}_t}}; \bar{\alpha}_t = \prod_{i=1}^{t}(1 - \beta_t).$$

We visualize the SNR evolution across time step $t > 0$ in Figure 12, where we can observe the SNR rapidly decays in the first 100 steps. According to previous studies in Arjovsky & Bottou (2017), injecting noise into the data distribution could smoothen the data distribution support and facilitate the GAN training. The SNR change in this interval indicates injecting noise in the level of $t \in [\![1, 100]\!]$ could bring in more significant improvement for the GAN training. When the step is greater than 200, the SNR is change is no longer significant and close to zero, which indicates the implicit model

---

[1]The pre-trained checkpoints are provided by: `https://github.com/pesser/pytorch_diffusion`

might not be too informative, though it is easier to train. Our experimental observations in Figure 3 also justify this conclusion: when training a GAN at $T_{\text{Trunc}} = 4$, the required number of iterations is similar to training it on clean data; by training the GAN model at $T_{\text{Trunc}} = 99$, the training of GAN is significantly facilitated. For $T_{\text{Trunc}} > 100$, we empirically examine to train a GAN and find it would converge faster than training the diffusion model for $t < T_{\text{Trunc}}$.

**Comparison of model efficiency**: In complement of the results in Table 1-2, we provide detailed model size and generation time on v100 GPU. The results are summarized in Table 7. We can see TDPM has an increasing in the total number of parameter, as it involves a discriminator to help train the implicit model, while its sampling efficiency is also obvious.

Table 7: Comparison of model size (the added parameters corresponds to the discriminator model in the training but not involved in the generation), and GPU time in generation.

| Resolution | 32×32 | | 64×64 | | 256×256 | |
|---|---|---|---|---|---|---|
| Model | Parameter | Time (s/image) | Parameter | Time (s/image) | Parameter | Time (s/image) |
| DDPM | 36M | 31.03 | 79M | 33.01 | 114M | 62.93 |
| TDPM, $T_{\text{trunc}}$=99 | 36M+20M | 3.13 | 79M+21M | 3.52 | 114M+24M | 6.65 |
| TDPM, $T_{\text{trunc}}$=49 | 36M+20M | 1.52 | 79M+21M | 1.55 | 114M+24M | 1.88 |
| TDPM, $T_{\text{trunc}}$=4 | 36M+20M | 0.16 | 79M+21M | 0.26 | 114M+24M | 0.65 |
| TDPM, $T_{\text{trunc}}$=0 | 36M+20M | 0.03 | 79M+21M | 0.05 | 114M+24M | 0.14 |

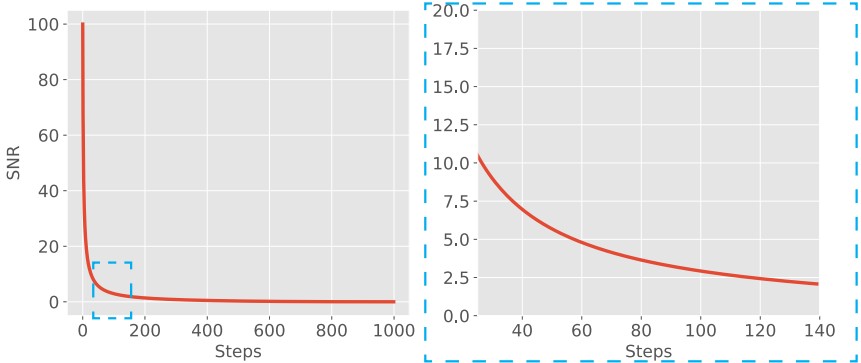

Figure 12: Signal-to-noise ratio evolution across different diffuse step $T$. The right sub-panel shows a zoomed-in SNR evolution in the range of [30, 140] steps.

### D.6 EXPERIMENTAL SETTINGS

#### D.6.1 MODEL ARCHITECTURE

**Generator:** Our generator structure strictly follows the U-Net structure (Ronneberger et al., 2015) used in DDPM, improved DDPM, and ADM (Ho et al., 2020; Nichol & Dhariwal, 2021; Dhariwal & Nichol, 2021), which consists of multiple ResNet blocks (He et al., 2016) with Attention blocks (Vaswani et al., 2017) injected in the bottleneck. Please refer to these paper for more details on the architecture.

A key difference between our model and previous diffusion models is that our model also train such U-Net as an extra implicit generator $G_\theta$ that takes a latent variable $\mathbf{z} \sim \mathcal{N}(\mathbf{0}, \mathbf{I})$ and a fixed time index $t = T_{\text{trunc}} + 1$ as input. However, this does not result in a difference in the generator architecture. We parameterize $G_\theta$ with the same U-Net architecture for simplicity and the time embedding $t = T_{\text{trunc}} + 1$ is specified to be trained with the implicit loss shown in (12) and (16). We have also tested to use all zero time embedding for $t = T_{\text{trunc}} + 1$ and found no clear differences.

For our results of TDPM+, the generator $G_\psi$ specifically takes a StyleGAN2 architecture Karras et al. (2020b) and there is no time-embedding in $G_\psi$. An increase of generator parameter appears caused by separating the implicit model and denoising U-Net. Note that the generator is trained with GAN loss and without specially designed adaptive augmentation in Karras et al. (2020a). For the detailed model architecture please refer to the corresponding paper or their Github repository: `https://github.com/NVlabs/stylegan2-ada-pytorch`.

**Discriminator:** Similar to Xiao et al. (2022), we adopt the discriminator architecture used in Karras et al. (2020b), but without the time step input. The discriminator discriminate $\mathbf{x}_{T_{\text{trunc}}}$ is from the diffused distribution $q(\mathbf{x}_{T_{\text{trunc}}})$ or implicit generative distribution $p_\theta(\mathbf{x}_{T_{\text{trunc}}})$. Please refer to Appendix C of Xiao et al. (2022) for the detailed design.

**Navigator:** Training with $\mathcal{L}_{T_{\text{trunc}}}^{\text{CT}}$ involves an extra module named navigator (Zheng & Zhou, 2021). We strictly follow the architecture used in Zheng & Zhou (2021), where the navigator is an MLP taking the pairwise feature distance as inputs. There is no time embedding used in the navigator as it is only used for the training at $t = T_{\text{Trunc}}$. The feature is extracted from the layer before the final scalar output. Please refer to their Appendix D for detailed information.

**Architecture for text-to-image experiments:** We adopt the 1.45B LDM model (Rombach et al., 2022) that is pretrained on LAION-400M dataset (Schuhmann et al., 2021). The LDM model consists of a U-Net KL-regularized autoencoder with downsampling-factor 8 (resolution 256 -> 32), a U-Net in the latent space, and a BERT (Devlin et al., 2018) text encoder transform raw text to a sequence of 1280-dimension embeddings. We only fine-tune the latent model in our experiments. In the training of the truncated part, the discriminator takes the first-half of the U-Net (downsampling backbone) with a linear predicting head on top of it.

**Architecture for toy experiments:** The generator uses an architecture stacked with 4 linear layers with 128 hidden units. Each intermediate layer is equipped with a time-embedding layer and follows softplus activation. The discriminator and navigator have the same architecture, without time-embedding layers, and using leakyReLU as the activation function.

### D.6.2 TRAINING CONFIGURATIONS

**Datasets:** We use CIFAR-10 (Krizhevsky et al., 2009), LSUN-bedroom, and LSUN-Church (Yu et al., 2015) datasets for unconditional generation in the main experiments. Additionally, we apply CelebA(Liu et al., 2015) and CelebA-HQ (Lee et al., 2020) for complementary justification. For text-to-image experiments, we use CUB-200 (Welinder et al., 2010) and MS-COCO (Lin et al., 2014). The images consist of $32 \times 32$ pixels for CIFAR-10. For the other datasets, we apply center-crop along the short edge and resize to the target resolution ($64 \times 64$ for CelebA; $256 \times 256$ for the others).

**Diffusion schedule:** For all datasets, we strictly follow the diffusion process used in our backbone models, and instantiate the truncated diffusion schedule by obtaining the first $T_{\text{Trunc}}$ diffusion rates $\{\beta_1, ..., \beta_{T_{\text{Trunc}}}\}$. For example, if our goal is to fit a model with NFE=50, to truncate the diffusion process used in Ho et al. (2020) ($\beta_1 = 10^{-4}$, $\beta_T = 0.02$, T=1000), we first initialize $\beta_1, \beta_2, ... \beta_{1000}$, and then taking the first 49 steps to complete the truncation.

**Optimization:** We train our models using the Adam optimizer (Kingma & Ba, 2015), where most of the hyperparameters match the setting in Xiao et al. (2022), and we slightly modify the generator learning rate to match the setting in Ho et al. (2020), as shown in Table 8.

We train our models using V100 GPUs, with CUDA 10.1, PyTorch 1.7.1. The training takes approximately 2 days on CIFAR-10 with 4 GPUs, and a week on CelebA-HQ and LSUN-Church with 8 GPUs.

Table 8: Optimization hyper-parameters.

|  | CIFAR10 | CelebA | CelebA-HQ | LSUN |
|---|---|---|---|---|
| Initial learning rate for discriminator | $10^{-4}$ | $10^{-4}$ | $10^{-4}$ | $10^{-4}$ |
| Initial learning rate for navigator (if applicable) | $10^{-4}$ | $10^{-4}$ | $10^{-4}$ | $10^{-4}$ |
| Initial learning rate for generator | $1 \times 10^{-5}$ | $1 \times 10^{-5}$ | $2 \times 10^{-5}$ | $2 \times 10^{-5}$ |
| Adam optimizer $\beta_1$ | 0.5 | 0.5 | 0.5 | 0.5 |
| Adam optimizer $\beta_2$ | 0.9 | 0.9 | 0.9 | 0.9 |
| EMA | 0.9999 | 0.9999 | 0.9999 | 0.9999 |
| Batch size | 128 | 128 | 64 | 64 |
| # of training iterations | 800k | 800k | 0.5M | 2.4M(bedroom)/1.2M(church) |
| # of GPUs | 4 | 8 | 8 | 8 |

For TDPM+, where we use StyleGAN2 generator as $G_\psi$, we directly use their original training hyper-parameters and train the model in parallel with the diffusion model. For TLDM, we set the base learning rate as $10^{-5}$ and the mini-batch size is set to 64. For the ImageNet1K-64×64 experiments, we use StyleGAN-XL generator as $G_\psi$ and strictly follow all the default training hyper-parameters. To simplify the implementation and save computation, instead of applying the default progressive growing pipeline $16 \times 16 \rightarrow 32 \times 32 \rightarrow 64 \times 64$, we directly train the implicit model on 64×64

images corrupted at $T_{\text{Trunc}}$. Without using the progressive growing pipeline, the result of StyleGAN-XL shown in Table 2 is clearly worse than the progressive one reported in their paper (FID 1.51). However, when used as the implicit model of TDPM, the final performance of TDPM becomes competitive with this result.

**Evaluation:** When evaluating the sampling time, we use models trained on CIFAR-10 and generate a batch of 128 samples. When evaluating the FID, and recall score, following the convention, we use 50k generated samples for CIFAR-10, LSUN-bedroom and LSUN-church, 30k samples for CelebA-HQ (since the CelebA HQ dataset contains only 30k samples), 30k samples for the text-to-image datasets. The recall scores are calculated with the recipe in Kynkäänniemi et al. (2019). In the sampling stage, we follow our backbone to apply the same guidance in the diffusion part ($t < T_{\text{Trunc}}$) if applicable. Specifically, for LDM backbone, we use classifier-free guidance (Ho & Salimans, 2022) with scale 1.5 and there are no DDIM steps for TDLM.

## D.7 ADDITIONAL RESULTS ON UNCONDITIONAL GENERATION

Table 9: Full comparison of unconditional generation on CIFAR-10. Models are grouped by the orders of sampling steps, with the best FID, and Recall in each group marked in bold. The TDPM+ results are produced using ADM and StyleGAN2 backbone. TDPM with NFE=1 is equivalent to training a GAN with the DDPM architecture as the generator.

| Model | NFE ↓ | FID ↓ | Recall ↑ |
|---|---|---|---|
| Improved DDPM (Nichol & Dhariwal, 2021) | 4000 | 2.90 | - |
| UDM (Kim et al., 2021) | 2000 | 2.33 | - |
| Likelihood SDE (Song et al., 2021a) | 2000 | 2.87 | - |
| Score SDE (VE) (Song et al., 2021b) | 2000 | **2.20** | 0.59 |
| Score SDE (VP) (Song et al., 2021b) | 2000 | 2.41 | 0.59 |
| NCSN (Song & Ermon, 2019) | 1000 | 25.3 | - |
| Adversarial DSM (Jolicoeur-Martineau et al., 2020) | 1000 | 6.10 | - |
| VDM (Kingma et al., 2021) | 1000 | 4.00 | |
| D3PMs (Austin et al., 2021) | 1000 | 7.34 | - |
| DiffuseVAE (Pandey et al., 2022), T=1000 | 1000 | 8.72 | - |
| DDPM (Ho et al., 2020) | 1000 | 3.21 | 0.57 |
| Recovery EBM (Gao et al., 2021) | 180 | 9.58 | - |
| Gotta Go Fast (Jolicoeur-Martineau et al., 2021) | 180 | 2.44 | - |
| LSGM (Vahdat et al., 2021) | 147 | **2.10** | **0.61** |
| Probability Flow (VP) (Song et al., 2021b) | 140 | 3.08 | 0.57 |
| DiffuseVAE (Pandey et al., 2022), T=100 | 100 | 11.71 | - |
| TDPM, $T_{\text{trunc}}$=99 (ours) | 100 | 2.97 | 0.57 |
| TDPM+, $T_{\text{trunc}}$=99 (ours) | 100 | 2.83 | 0.58 |
| FastDDPM, T=50 (Kong & Ping, 2021) | 50 | 3.41 | 0.56 |
| DDIM, T=50 (Song et al., 2020) | 50 | 4.67 | 0.53 |
| SNGAN+DGflow (Ansari et al., 2021) | 25 | 9.62 | 0.48 |
| TDPM, $T_{\text{trunc}}$=49 (ours) | 50 | **3.11** | 0.57 |
| TDPM+, $T_{\text{trunc}}$=49 (ours) | 50 | **2.96** | **0.58** |
| Progressive distiallation (Salimans & Ho, 2022) | 8 | **2.57** | - |
| Denoising Diffusion GAN (Xiao et al., 2022), T=8 | 8 | 4.36 | 0.56 |
| Progressive distiallation (Salimans & Ho, 2022) | 4 | **3.00** | - |
| Denoising Diffusion GAN (Xiao et al., 2022), T=4 | 4 | 3.75 | **0.57** |
| TDPM, $T_{\text{trunc}}$=4 (ours) | 5 | 3.51 | 0.55 |
| TDPM+, $T_{\text{trunc}}$=4 (ours) | 5 | 3.17 | **0.57** |
| Progressive distiallation (Salimans & Ho, 2022) | 2 | 4.51 | - |
| Denoising Diffusion GAN (Xiao et al., 2022), T=2 | 2 | 4.08 | 0.54 |
| TDPM, $T_{\text{trunc}}$=1 (ours) | 2 | 4.47 | 0.53 |
| TDPM+, $T_{\text{trunc}}$=1 (ours) | 2 | **3.86** | **0.56** |
| DDPM Distillation (Luhman & Luhman, 2021) | 1 | 9.36 | **0.51** |
| SNGAN (Miyato et al., 2018) | 1 | 21.7 | 0.44 |
| AutoGAN (Gong et al., 2019) | 1 | 12.4 | 0.46 |
| TransGAN (Jiang et al., 2021) | 1 | 9.26 | - |
| StyleGAN2 w/o ADA (Karras et al., 2020a) | 1 | 8.32 | 0.41 |
| StyleGAN2 w/ ADA (Karras et al., 2020a) | 1 | **2.92** | 0.46 |
| StyleGAN2 w/ Diffaug (Zhao et al., 2020) | 1 | 5.79 | 0.42 |
| Progressive distiallation (Salimans & Ho, 2022) | 1 | 9.12 | - |
| Denoising Diffusion GAN (Xiao et al., 2022), T=1 | 1 | 14.6 | 0.19 |
| TDPM, $T_{\text{trunc}}$=0 (ours) | 1 | 7.34 | 0.46 |

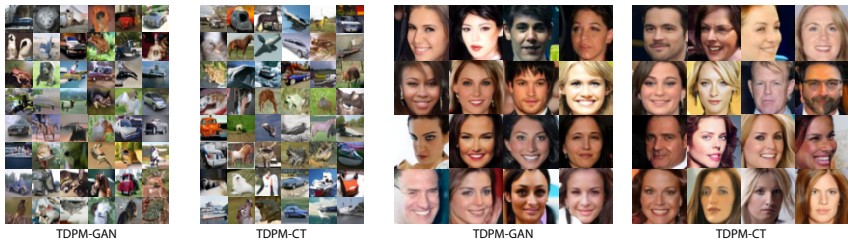

Figure 13: Qualitative results on CIFAR-10 and CelebA ($64 \times 64$).

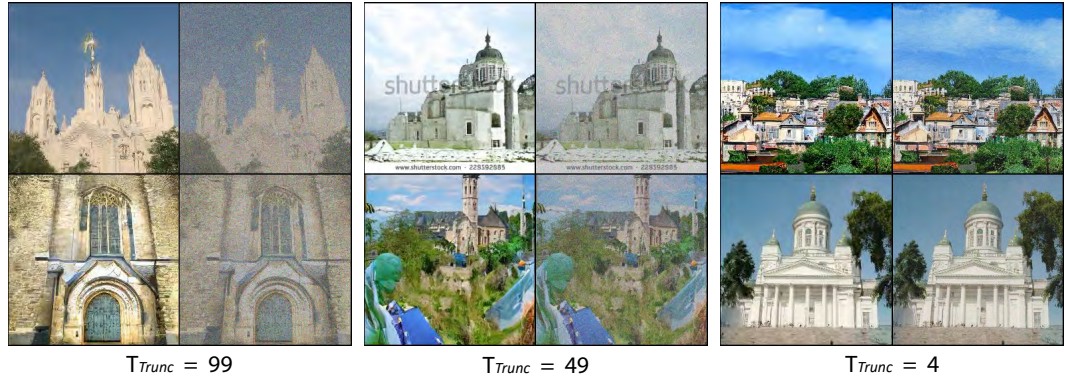

Figure 14: Qualitative results of TDPM on LSUN-Church ($256 \times 256$), with $T_{\text{trunc}} = 99$, 49, and 4. Note NFE $= T_{\text{trunc}} + 1$ in TDPM. Each group presents generated samples from $p_\theta(\mathbf{x}_0)$ (left) and $p_\theta(\mathbf{x}_{T_{\text{trunc}}})$ (right).

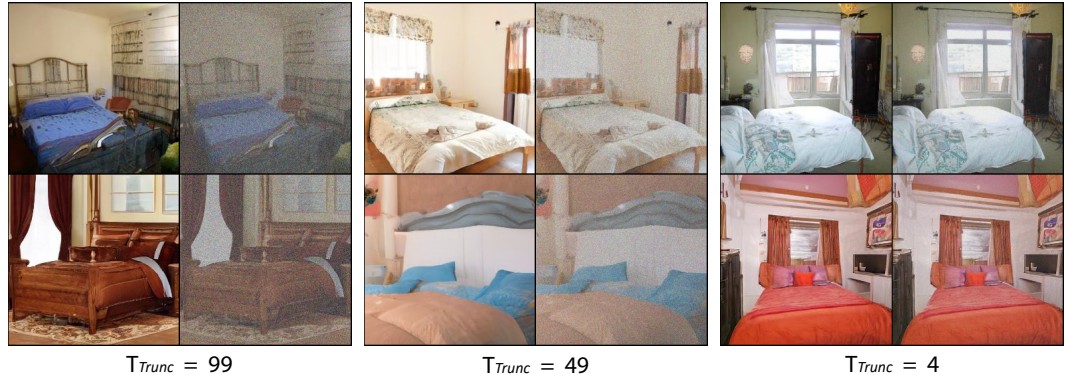

Figure 15: Analogous qualitative results to Figure 14 on LSUN-Bedroom. Produced by TDPM.

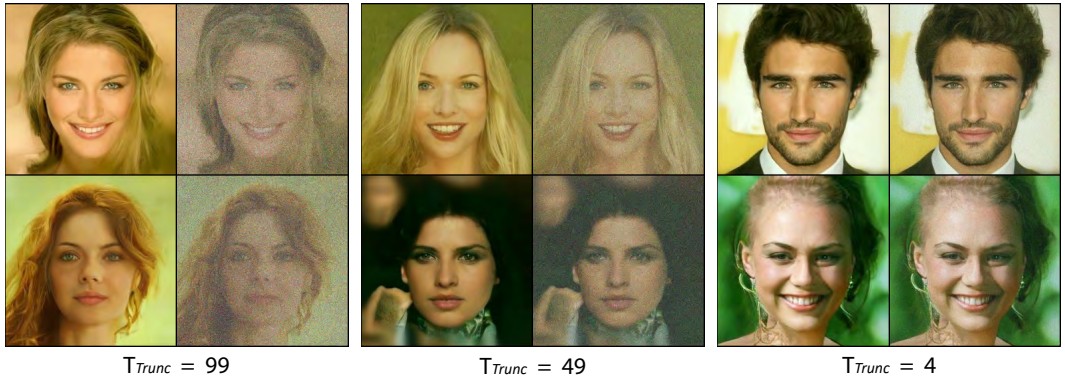

Figure 16: Analogous qualitative results to Figure 14 on CelebA-HQ. Produced by TDPM.

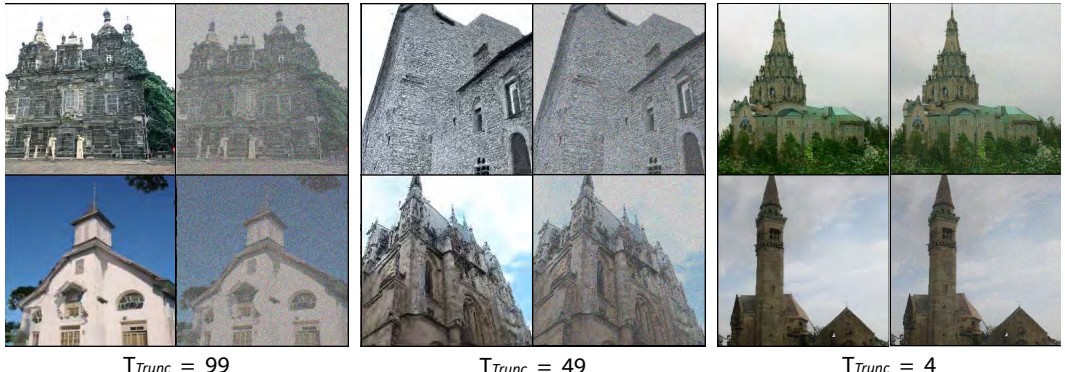

Figure 17: Analogous qualitative results to Figure 14 on LSUN-Church. Produced by TDPM-CT.

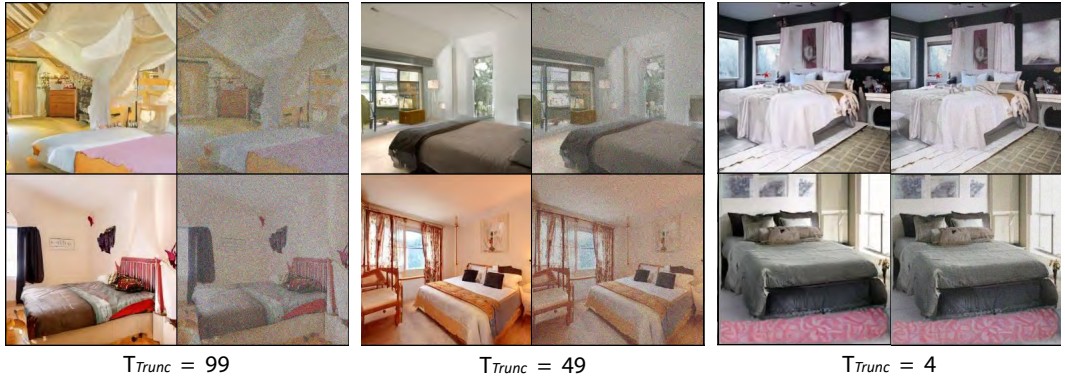

Figure 18: Analogous qualitative results to Figure 14 on LSUN-Bedroom. Produced by TDPM-CT.

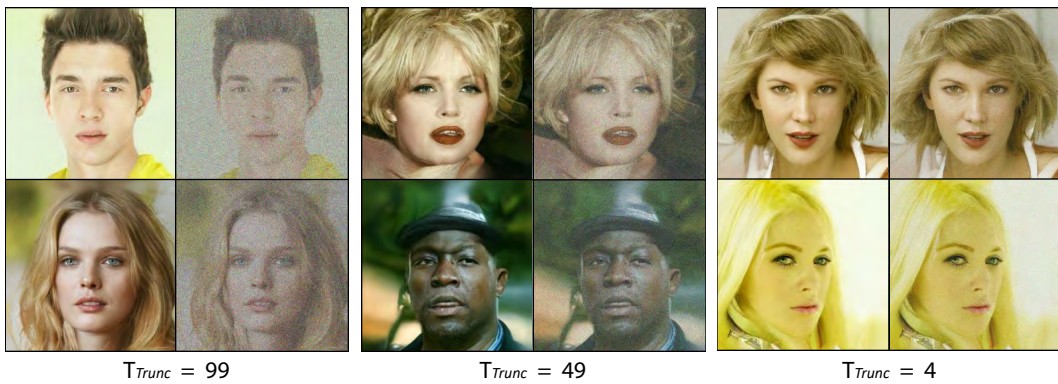

Figure 19: Analogous qualitative results to Figure 14 on CelebA-HQ. Produced by TDPM-CT.

### D.8 ADDITIONAL RESULTS ON TEXT-TO-IMAGE GENERATION

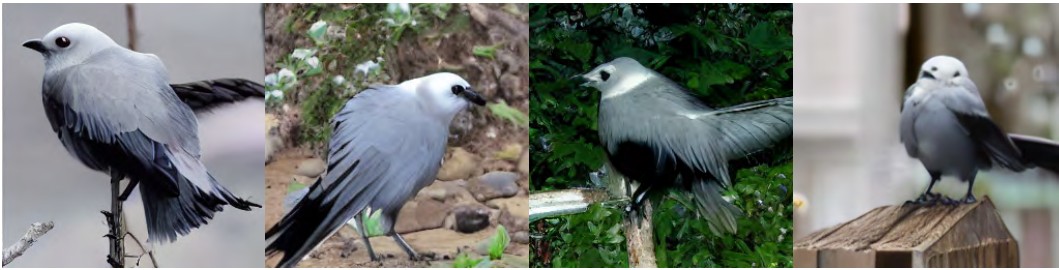

A white and gray bird with black wings.

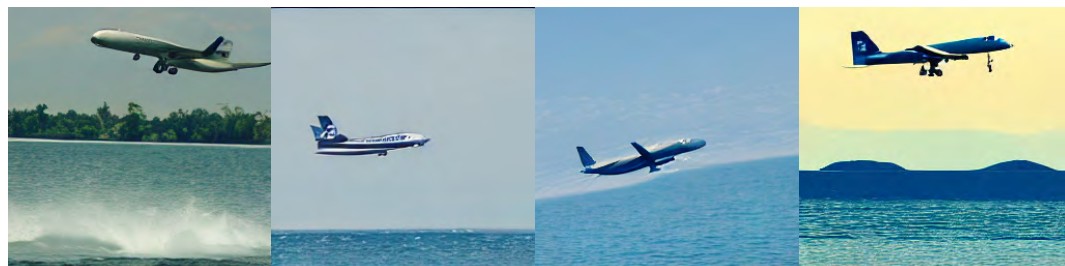

An airplan flying over a body of water.

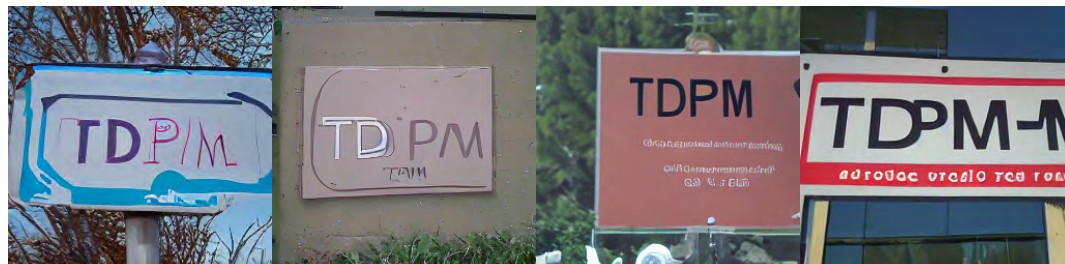

A sign reads "TDPM".

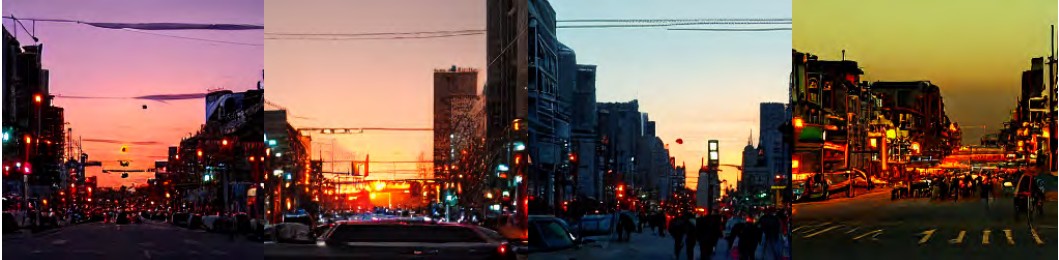

Busy city street at dusk with sun setting.

Figure 20: Additional text-to-image generation results with different text prompt, produced by TLDM with $T_{\text{trunc}} = 49$.

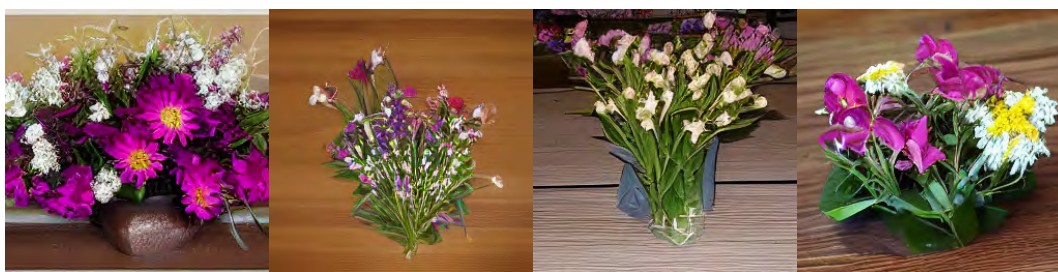

A cluster of flower on the wooden table.

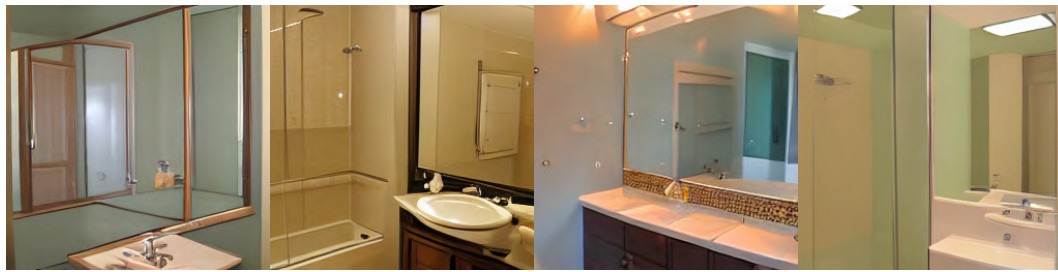

The bathroom has a big mirror.

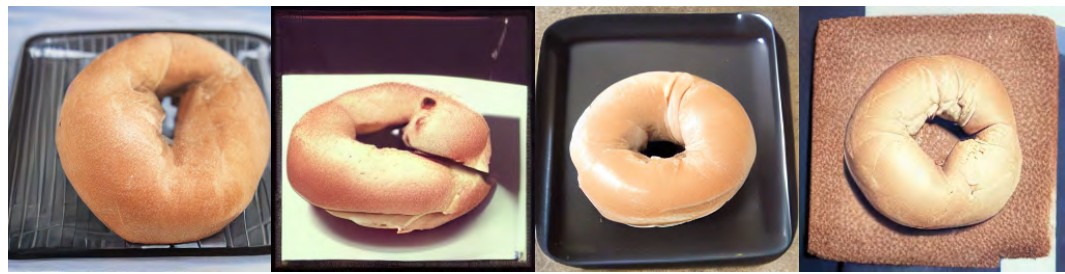

The bagel is put in a squre plate.

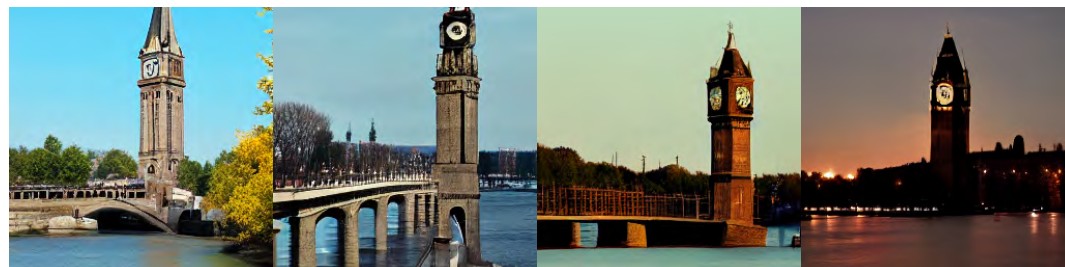

A clock tower near the river.

Figure 21: Additional text-to-image generation results with different text prompt, produced by TLDM with $T_{\text{trunc}} = 4$.

