# OpenReview forum: "Truncated Diffusion Probabilistic Models and Diffusion-based Adversarial Auto-Encoders"
_ICLR.cc/2023/Conference — ICLR 2023 poster_

### Official Review · Reviewer_MGCM · 2022-10-22

**Confidence:** 5
**Correctness:** 4
**Technical Novelty And Significance:** 3
**Empirical Novelty And Significance:** 3
**Recommendation:** 8

**Clarity, Quality, Novelty And Reproducibility:**

The paper is well written and the method is clearly presented. The idea is novel, to the best of my knowledge. The method is straightforward to implement.

**Strength And Weaknesses:**

Strength:

1. The idea is quite interesting. Previously people have observe when sampling from DDPM and do denoising for 1000 steps, the steps close to clean sample $x_0$ are more important to steps close to white noise $x_1000$. Some noise scheduling techniques are designed based on this observation, but this method takes an interesting approach to utilize this observation. It simply by-pass the early steps with a single step model such as GAN, and start sampling from diffusion models from a truncated time step.

2. Although I feel like it is a combination of denoising diffusion GAN by Xiao et al. and traditional DDPM, but the benefit of such a combination is clear. Basically, the model can be interpreted as applying one step of denoising diffusion GAN, and using DDPM for remaining steps. However, it obtains better results, and it maintains the nice property of DDPM (while DDGAN is more like a GAN).

3. Strong experimental results. It is nice to see we can even surpass the sample quality of DDPM with fewer sampling steps, while many previous speed-up methods have a degrade in sample quality.

Weakness:
1. Just like DDGAN, it loses some property of DDPM such as likelihood estimation, due to the GAN.

2. Not necessarily a weakness, but I have one question. I am interested in the results of truncated at 4 steps. If you truncated only at 4 steps out of 1000 steps, is it almost equivalent to training a GAN (or DDGAN with T=1)? Because at t=4, the noisy sample $x_4$ is almost clean, and you need to train a GAN to map white noise to $x_4$. However, from DDGAN paper, it seems like DDGAN at $T=1$ does not work very well. I would be interested in knowing more details on this.

**Summary Of The Paper:**

The paper introduces truncated diffusion models, which truncates the forward process at certain timesteps so that when sampling, only a small portion of the reverse process is needed, which saves sampling time. To map noise to the noisy data at the truncated timestep, a conditional GAN is trained. The paper also builds up the relation between truncated diffusion models and AAE. Experimental results show that the model can generate similar or better samples than DDPM with a small number of NFEs.

**Summary Of The Review:**

I think this paper propose a novel and effective idea of speed-up sampling from diffusion models. The empirical results are also promising. Therefore, I think this is a good submission to be accepted.

---

> ### Author Response · Authors · 2022-11-18
> **Response to Reviewer MGCM**
>
> We appreciate your recognition and positive and great feedback on our work
>
> > It loses some property of DDPM such as likelihood estimation, due to the GAN.
>
> Although likelihood evaluations via ODE may no longer be suitable, they could still be viable using other techniques. For example, TDPM can be formulated as a semi-implicit distribution, whose first stochastic layer $p(x_0|x_1)$ is an analytic Gaussian distribution and the subsequent stochastic layers together form an implicit distribution for $x_1$. This implicit distribution on $x_1$ has no analytic density function, but is simple to sample from. We note that the data likelihood under a semi-implicit distribution could be estimated using Monte Carlo estimation (combining Gaussian likelihoods $p(x_0|x_1^{(s)})$ over iid random samples $x_1^{(s)}$ from that implicit distribution). We defer likelihood evaluations under TDPM to future work.
>
> -------------------------------
>
> > Discussions on $T_{Trunc}=4$
>
> As you note that DDGAN does not do well when T=1 ($T_\text{Trunc} = 0$ for us), we also see that using GANs with U-Net backbone on clean images gives worse results than when $T_\text{Trunc}$ is larger (e.g., 4, 49), as Table 1-2 and Fig. 4 show. We think that some hard-to-train information in the images may be filtered out by adding a little noise at $T_\text{Trunc}=4$, even though the images still look clean. One paper we cited (Arjovsky et al., 2017) also says that adding noise at the right level can help GAN training and improve the results.

---

### Official Review · Reviewer_YWr1 · 2022-10-24

**Confidence:** 3
**Correctness:** 4
**Technical Novelty And Significance:** 3
**Empirical Novelty And Significance:** 3
**Recommendation:** 8

**Clarity, Quality, Novelty And Reproducibility:**

The paper is easy to understand and the quality of the paper is generally good. I appreciate a lot of additional information in the supplementary materials like the toy example, alternative generator, etc.

I think the reproducibility is also pretty good, considering that this is simply adding GAN to the (start) end of the (reverse) diffusion process which is easy to implement. Hyper-parameters used is also mentioned in the supp.

As for the novelty, as mentioned in the related work, the idea of trying to speed up diffusion has been studied quite a lot, though I believe not exactly as what has been proposed here. Most similar I think is this recent work "Accelerating Diffusion Models via Early Stop of the
Diffusion Process" from Lyu et al which propose similar idea of sampling from implicit distribution learned from GAN or VAE, though they seem to be using pre-trained generator instead of training them together.

**Strength And Weaknesses:**

Strength

-well motivated approach

-well written paper

-extensive analysis and discussion


Weakness

-the step to truncate T_trunc need to be decided at training time, and it's not clear what is the best T_trunc to set.

-experiments could have been better, with more diverse and higher quality datasets. The main paper only shows cifar-10, LSUN-church, and LSUN-Bedroom. Something like ImageNet for example, could be more convincing.

-introducing adversarial aspect also means potential training difficulty that comes with GAN




**Summary Of The Paper:**

The authors propose TDPM, a truncated diffusion probabilistic model that essentially skips the diffusion steps by truncating the start/end of the process, stopping at an implicit non-gaussian distribution which can be sampled from another generative model. The goal is to reduce the number of steps without compromising image quality. The authors also related the proposed TDPM to an adversarial autoencoder AAE in the same way as how DDPM is sort of a VAE. Given T_trunc as the end point instead of T where T_trunc is a lot less than T, the discriminator is thus trying to discriminate between sample from this implicit distribution at T_trunc and from the generator. Interestingly, this generator can even be the same as the denoising network (by specifying appropriate t). The results seem convincing,

**Summary Of The Review:**

Overall I really like the paper. The problem of speeding up diffusion is an important one and the paper does a good job of convincing the readers the validity of the proposed approach. The results seem to support the claim that the proposed method can speed up the process without sacrificing image quality. With that said, based on the fact that t_trunc=4 already provide quite good results already, I feel like it is mostly shifting toward being a GAN than a diffusion. Still, table 1 and 2 suggest that having diffusion even as small as 4 steps does provide benefit over pure GAN.

Others:

L_CT in section 3.5 is not defined in the main paper, even though it does in the supplementary material.

Is there any particular reasons for the choice of 4, 49 and 99 T_trunc?

---

> ### Author Response · Authors · 2022-11-18
> **Response to Reviewer YWr1**
>
> Thank you for your great feedback and acknowledgment of our work.
>
> > How to determine $T_{trunc}$ in training?
>
> Our observations over a diverse set of datasets consistently show that when $T_\text{Trunc}$ is as large as 99, there is negligible performance degradation; when we make $T_\text{Trunc}$ as small as 4, TDPM suffers minor drops in performance. In other words, there are no drastic performance differences between the $T_\text{Trunc}$’s belonging to $ [4,5,\ldots, 99]$. Our recommendation for an end user would be choosing an appropriate $T_\text{Trunc}\in [4,5,\ldots, 99]$ according to their computational constraint.
>
> ------------------------------
>
>
> > Experiments on ImageNet
>
> We have added the results of ImageNet in the revision. Please refer to our Table 3 and Section 4.2, which show that TDPM performs well on ImageNet and demonstrates a performance improvement over the backbone.
>
> ------------------------------
>
> > Introducing the adversarial aspect also means potential training difficulty
>
> We find that properly choosing $T_\text{Trunc}$ could relax the difficulty of training a GAN, which is also suggested in Arjovsky et al., 2017, cited in our paper. In such case, we believe GAN and diffusion models could help each other to overcome the training difficulty of each counterpart. As shown in Fig 3, at $T_\text{Trunc}=100$, the training is more efficient than training a vanilla GAN or a diffusion model.
>
> ------------------------------
>
> > L_CT in section 3.5 is not defined in the main paper, even though it does in the supplementary material.
>
> In the revision, we move all parts related to the CT into supplementary.
>
> ------------------------------
>
> > Are there any particular reasons for the choice of 4, 49 and 99 T_trunc?
>
> In TDPM, the first step through the implicit model takes one NFE, and hence the total number of NFE is $T_\text{Trunc}+1$. We choose $T_\text{Trunc}=$4,49, and 99 to make $\text{NFE}=$5,50, and 100, allowing us to make fair comparisons with baselines under these specific NFEs.

---

### Official Review · Reviewer_vJWX · 2022-10-25

**Confidence:** 4
**Correctness:** 3
**Technical Novelty And Significance:** 2
**Empirical Novelty And Significance:** 2
**Recommendation:** 5

**Clarity, Quality, Novelty And Reproducibility:**

* This manuscript is generally well-written. I’m able to follow many technical details to understand the proposed method. There are some minor issues, summarized in the section above.

* In terms of novelty, this is not the first work to combine different types of generative models, since DD-GAN also shows the potential that considering implicit generative models in the diffusion process reduces the forward or reverse steps.

* In terms of reproducibility, I didn’t find any unusual components to implement the proposed method. So, the numbers introduced in the paper would be reproduced.

**Strength And Weaknesses:**

**Strengths**:
* The problem tackled in this paper is important both in academic and practical scenarios. And, the proposed method, combining diffusion process and implicit generative models, is also reasonable, since the hidden-noisy distribution after truncated diffusion process, seems to be a quiet unimodal distribution, which will be well estimated by GANs.
* Experiments on unconditional and text-conditional generation tasks show that TDPM reasonably performs well even in the case of a very limited number of reverse steps.

**Weaknesses**:
* In my opinion, the main weakness of this work is insufficient comparison to previous work. To improve the sampling speed in the DDPM framework, there are several methods published in ICLR last year. This manuscript briefly discusses the limitation of DD-GAN and progressive distillation, but none of them are empirically compared to TDPM. So, this makes it difficult to evaluate the real value of TDPM in practical scenarios.

**Detailed comments**

As shown in Eq.(14), the authors try to train a network by jointly optimizing the denoising objective and adversarial loss. This may hurt the stability of optimization for large-scale diffusion models. Instead of joint optimization, what about using this two-phase approach? First, the standard denoising objective is used to pre-train the diffusion model. Then, we truncate the diffusion process and train GANs to match prior to aggregated posterior.

I failed to find insights from recasting TDPM as an adversarial auto-encoder, since this reinterpretation seems to be fairly straightforward. Honestly, even if the derivation is straightforward, it would be great to give many insights through the reinterpretation.

It would be much better to compare the proposed method with progressive distillation or DD-GAN. In my understanding, DD-GAN was not proven to be working well on ImageNet-scale datasets, but progressive distillation works well on many large-scale datasets.

There are some minor comments to the manuscript:
* I couldn’t find \mathcal{L}_{\textrm{T}_{\textrm{trunc}}^{\textrm{CT}} in the main section. It first appears in the appendix.
* Figure 5 shows some samples having some artifacts caused by watermarks in the training set. It would be better to include clean samples.

**Summary Of The Paper:**

This paper aims to improve sampling speed of diffusion-based generative models by minimizing the number of reverse steps. Instead of using a distillation technique, this paper truncates the diffusion process, stopping adding noise to samples before the samples become pure white noise. Then, an implicit generative model is employed to model this hidden-noisy distribution for sampling. Experiments on unconditional and text-conditional generation tasks show that the proposed method performs reasonably well in case of a limited number of reverse steps.

**Summary Of The Review:**

Though the idea combined with likelihood-based and implicit generative models is always appealing to me, the insufficient empirical justification makes me hesitate to accept this work. if the manuscript is updated in the revision phase, I reconsider my initial evaluation.

---

> ### Author Response · Authors · 2022-11-18
> **Response to Review vJWX**
>
> Thank you for your comments, below are our responses regarding your concerns and questions.
>
> > More comparisons with DDGAN and Progressive distillation. Adding ImageNet results.
>
> Thanks for the suggestion, we have added the suggested comparisons into Tables 1-2 and created a new Table in the supplementary (Table 9) to add more baselines.
> We have also added the results of ImageNet, mainly comparing with our backbones, ADM and StyleGAN-XL, in Table 3.
>
> We could not find the exact FID of Progressive Distillation on ImageNet-64. We estimate their values from their Figure 4 and show them in the table below for easier comparison.  We can observe TDPM surpasses both backbone models and Progressive Distillation.
>
>
>
>
> | Method                      | NFE  | FID           | Recall         |
> |-----------------------------|------|---------------|----------------|
> | ADM                         | 1000 | 2.07          | **0.63**  |
> | TDPM+ ($T_\text{Trunc}$=99) | 100  | **1.62** | **0.63** |
> | TDPM+ ($T_\text{Trunc}$=49) | 50   | 1.77          | 0.58        |
> | TDPM+ ($T_\text{Trunc}$=4)  | 5    | 1.92          | 0.53           |
> | StyleGAN-XL (wo PG)         | 1    | 3.54          | 0.51           |
> | PG Distill-8steps           | 8    | $\approx$2.80 | -              |
> | PG Distill-4steps           | 4    | $\approx$3.50 | -              |
> | PG Distill-2steps           | 2    | $\approx$7.00 | -              |
>
> Moreover, to further compare TDPM with Progressive Distillation, we note that the performance of Progressive Distillation on larger-scale diverse datasets like ImageNet shows a significant gap to the teacher model, even using more sampling steps, which can be observed in Figure 4 of the Progressive Distillation paper. TDPM does not have this problem, and it can further improve the diffusion backbone by using a stronger implicit model.
>
> -----------------
>
> > What about using this two-phase approach? First, the standard denoising objective is used to pre-train the diffusion model. Then, we truncate the diffusion process and train GANs to match prior to aggregated posterior.
>
>
> This two-phase approach resembles our TDPM+ configuration, but differs in the range of steps for which the diffusion model is pre-trained: {1, …, T} or {1,…, $T_\text{Trunc}$}. The former option takes more time for the diffusion model to converge, as shown in our figure 3. We think that TDPM+ is preferable in this case. If a pre-trained diffusion model is already available, TDPM+ can benefit from it and become equivalent to the former option, and we only need to train the implicit generator to match the aggregated posterior. We report the results of using a pre-trained DDPM/improved-DDPM in Table 5 in the Appendix. We observe that TDPM, which combines a pre-trained diffusion model for $t<{T_\text{Trunc}}$ and a trainable implicit model at $t={T_\text{Trunc}}$, also achieves competitive performance compared with TDPM trained end-to-end.
>
> -----------------
>
> > Relation between TDPM and AAE
>
> The relation between TDPM and AAE helps readers to understand the distinction between TDPM and DDPM, as well as the distinction between TDPM and GAN models (e.g. DDGAN).
>
> The difference between DDPM vs TDPM is analogous to the difference between VAE and AAE, which differs in matching a variational distribution and matching an aggregated posterior distribution with the prior distribution. We no longer require the prior distribution at $T_\text{Trunc}$ to be analytic and allow it to take an implicit distribution. More detailed differences can be found in Makhzani et al., 2015.
> Compared with GANs, TDPM only leverages the implicit distribution to match the aggregated posterior instead of matching the data distribution. We still rely on the diffusion model for $t < T_\text{Trunc}$ to capture the data distribution, which preserves the properties of likelihood-based methods.
>
> To see how TDPM differs from DDPM and GAN models (e.g. DDGAN), readers can compare TDPM and AAE, which have a similar relation. Both TDPM and AAE match the prior distribution with an aggregated posterior distribution of the latent variables, instead of with a variational distribution as used in DDPM and VAE. This way, the latent distribution at $T_{Trunc}$ can be implicit and more flexible, without needing an analytic form. Makhzani et al., 2015 explain this difference in more detail.
>
> TDPM also differs from GANs in how it uses implicit distribution. TDPM only uses it to match the combined latent distribution, not the data distribution. For $t < T_\text{Trunc}$, TDPM still uses the diffusion model to model the data distribution, which preserves the advantages of likelihood-based methods.
>
> -----------------
>
> > Improving the presentation
>
> We have moved $L_\text{CT}$ to the correct section in the appendix and improved the quality of Figure 5.

---

### Official Review · Reviewer_6UUu · 2022-10-25

**Confidence:** 4
**Correctness:** 4
**Technical Novelty And Significance:** 2
**Empirical Novelty And Significance:** 3
**Recommendation:** 6

**Clarity, Quality, Novelty And Reproducibility:**

The central idea in this paper is finding a suitable middle ground between GANs and diffusion models (DMs). In this sense, the novelty of the paper is somewhat limited, as a very similar concept has been explored in [1].

However, quantitatively speaking, the model performs better than [1] in terms of both FID and Recall, and also better than many other competing methods that exist in the space of fast diffusion models (e.g. DDIM, FastDDPM, Distilled Diffusion). Therefore, it may suggest that refining GAN predictions with a score may be more effective than using GANs throughout.

[1] Xiao, Z., Kreis, K. and Vahdat, A., 2021. Tackling the generative learning trilemma with denoising diffusion gans. arXiv preprint arXiv:2112.07804.

**Strength And Weaknesses:**

Strengths:
- TDPMs boast shorter diffusion times.
- There appears to be little sacrifice in sample quality (in terms of FID) and mode coverage (in terms of recall).

Weaknesses:
- The ability to perform likelihood evaluations via the deterministic ODE framework is lost, as the $p_{T_{Trunc}}$ model is now implicit.
- It is unclear how to choose $T_{Trunc}$, and how to control the trade-off between speed versus sample quality and mode coverage.

**Summary Of The Paper:**

For faster synthesis in diffusion models, this work proposes a sampling procedure that simulates a truncated diffusion process. For example, if a standard diffusion model simulates a diffusion process for $t \in [0, T]$, its truncated variant (which the authors call the TDPM) simulates $t \in [0, T_{Trunc}]$, where $T_{Trunc} < T$. To then sample from this process, $x$ is drawn from $p_{T_{Trunc}}$ (rather than $p_T$), and then the diffusion is reversed to $t=0$ as in a standard diffusion model. Of course, $p_{T_{Trunc}}$ is usually not a simple distribution anymore and cannot be trivially sampled from; the authors thus choose to model it implicitly in a GAN-like fashion.

**Summary Of The Review:**

The authors propose a hybrid GAN/diffusion model that trades off between the strengths and weaknesses of GANs and diffusion models. While the basic motivating principle has already been explored, I am on the fence about whether its simplicity and execution may warrant greater attention and discussion.

---

> ### Author Response · Authors · 2022-11-18
> **Response to Reviewer 6UUu**
>
> We appreciate your feedback and address your queries as follows:
>
> > The ability to perform likelihood evaluations via the deterministic ODE framework is lost
>
>
> Although likelihood evaluations via ODE may no longer be suitable, they could still be viable using other techniques. For example, TDPM can be formulated as a semi-implicit distribution, whose first stochastic layer $p(x_0|x_1)$ is an analytic Gaussian distribution and the subsequent stochastic layers together form an implicit distribution for $x_1$. This implicit distribution on $x_1$ has no analytic density function, but is simple to sample from. We note that the data likelihood under a semi-implicit distribution could be estimated using Monte Carlo estimation (combining Gaussian likelihoods $p(x_0|x_1^{(s)})$ over iid random samples $x_1^{(s)}$ from that implicit distribution). We defer likelihood evaluations under TDPM to future work.
>
> -----------------
>
>  > It is unclear how to choose $T_{Trunc}$  and how to control the trade-off between speed versus sample quality and mode coverage.
>
>
>
> Our experiments show that the performance of TDPM is robust to the choice of $T_{Trunc}$ within the range of $ [4,5,\ldots, 99]$. When $T_{Trunc}$ is 99, the performance is almost optimal; when $T_{Trunc}$ is 4, the performance is slightly worse. In other words, the performance differences between different $T_{Trunc}$’s in this range are minor. Our suggestion for a user would be to select a suitable $T_{Trunc}\in [4,5,\ldots, 99]$ based on their computational budget.
>
> -----------------
>
>
> > The central idea of this paper is to find a suitable middle ground between GANs and diffusion models (DMs). In this sense, the novelty of the paper is somewhat limited, as a very similar concept has been explored in DDGAN.
>
> We would like to clarify that TDPM differs from DDGAN in several perspectives:
> DDPM can be viewed as a hierarchical VAE with a fixed encoder, while TDPM can be viewed as a hierarchical AAE with a fixed encoder. DDGAN trains each diffusion step adversarially and behaves more like a GAN.
> Mathematically, DDGAN replaces $\sum_{t=1}^{T} \mathcal L_{t-1}$ with an adversarial version, while TDPM modifies $\mathcal L_T$ of the DDPM loss $\mathcal \sum_{t=1}^{T} \mathcal L_{t-1} + \mathcal L_{T}$. As a result, DDGAN deviates significantly from DDPM, while TDPM still retains some of the properties of the diffusion models.
>
> The results of DDGAN indicate $T=4$ is preferred, and a larger $T$ often degrades its performance. In TDPM we find that a larger $T_{Trunc}$ is helpful for the GAN training, and choosing a $T_{Trunc}$ around 49 or 99 can benefit both the diffusion model and GAN.

---

### Author Response · Authors · 2022-11-18
**Summary of revisions**

We thank all reviewers for their helpful feedback and suggestions. We have revised the paper carefully and highlighted our changes in blue. The main points of revision are summarized below for your convenience:

- We have included the results on ImageNet1K (64x64 resolution) in Table 3. Figure 2 is updated to show ImageNet-64 generations instead of CIFAR-10 generations. . The corresponding description is added in Section 4.2. The experiment details of TDPM on Image-net are provided in Appendix D.7.

- We have compared our method with DDGAN (Xiao et al. 2021) in Tables 1, 2, and 9. We have added the results of progressive distillation (Salimans et al. 2022) and several additional baselines in Table 9.

- We have added in Table 5 the results of TDPM that uses a two-step procedure: first pretraining a DDPM and then training an implicit model at $T_\text{Trunc}$ of this pre-trained DDPM, combining them as a TDPM.

---

### Decision · Program_Chairs · 2023-01-20

**Decision:**

Accept: poster

**Justification For Why Not Higher Score:**

Either poster or spotlight. Although all reviews were positive, and this is a solid paper, not all reviewers were highly convinced.

**Justification For Why Not Lower Score:**

This is a solid paper.

**Metareview: Summary, Strengths And Weaknesses:**

Ratings: 6/5/8/8
Confidence: 4/4/3/5
Recommendation: Accept

This paper proposes a technique, TDPM, for faster sampling from diffusion model. In particular, the authors propose to truncate the forward diffusion model at the end, such that the distribution at the last timestep still has some structure left. This left-over structure is modeled through a GAN-like fashion, with an adversarial autoencoder.

Strengths:
- faster sampling
- little sacrifice in image quality (in terms of FID) and mode coverage (in terms of recall).
- complementary to distillation approaches.

There seem to be few significant weaknesses. Some reviewers asked for ImageNet experiments, which are added later. Unfortunately there was no discussion later on this.

Based on the positive reviews, and the fact that the paper was even improved after that with more experiments and improved organization, I recommend to accept the paper.


**Note From Pc:**

if the above contains the word "oral" or "spotlight" please see: "oral" presentation means -> notable-top-5% and "spotlight" means -> notable-top-25%. As stated in our emails, we are disassociating presentation type from AC recommendations